# Blackbox Model Provenance via Palimpsestic Membership Inference

**Rohith Kuditipudi**[*]
rohithk@stanford.edu

**Jing Huang**[*]
hij@stanford.edu

**Sally Zhu**[*]
salzhu@stanford.edu

**Diyi Yang**[†]
diyiy@cs.stanford.edu

**Christopher Potts**[†]
cgpotts@stanford.edu

**Percy Liang**[†]
psl@stanford.edu

Department of Computer Science
Stanford University

## Abstract

Suppose Alice trains an open-weight language model and Bob uses a blackbox derivative of Alice's model to produce text. Can Alice prove that Bob is using her model, either by querying Bob's derivative model (query setting) or from the text alone (observational setting)? We formulate this question as an independence testing problem—in which the null hypothesis is that Bob's model or text is independent of Alice's randomized training run—and investigate it through the lens of *palimpsestic memorization* in language models: models are more likely to memorize data seen later in training, so we can test whether Bob is using Alice's model using test statistics that capture correlation between Bob's model or text and the ordering of training examples in Alice's training run. If Alice has randomly shuffled her training data, then any significant correlation amounts to exactly quantifiable statistical evidence against the null hypothesis, regardless of the composition of Alice's training data. In the query setting, we directly estimate (via prompting) the likelihood Bob's model gives to Alice's training examples and their training order; we correlate the likelihoods of over 40 fine-tunes of various Pythia and OLMo base models ranging from 1B to 12B parameters with the base model's training data order, achieving a p-value on the order of at most $1 \times 10^{-8}$ in all but six cases. In the observational setting, we try two approaches based on estimating 1) the likelihood of Bob's text overlapping with spans of Alice's training examples and 2) the likelihood of Bob's text with respect to different versions of Alice's model we obtain by repeating the last phase (e.g., 1%) of her training run on reshuffled data. The second approach can reliably distinguish Bob's text from as little as a few hundred tokens; the first does not involve any retraining but requires many more tokens (several hundred thousand) to achieve high power.

## 1 Introduction

**Definition 1.** A *palimpsest* is a "writing material (such as a parchment or tablet) used one or more times after earlier writing has been erased" [1].

---

[*]Equal contribution.
[†]Equal advising.

39th Conference on Neural Information Processing Systems (NeurIPS 2025).

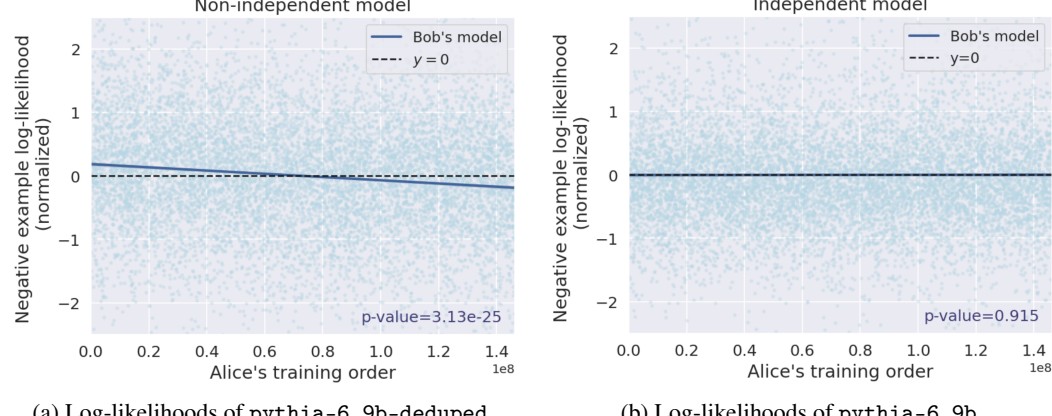

(a) Log-likelihoods of `pythia-6.9b-deduped`.  (b) Log-likelihoods of `pythia-6.9b`.

Figure 1: We regress the negative log-likelihoods of `pythia-6.9b-deduped` and `pythia-6.9b` on `pythia-6.9b-deduped` training examples against its training order, which is independent of `pythia-6.9b` training order. Though the log-likelihoods of individual examples are noisy, the overall trend is clear over many examples: `pythia-6.9b-deduped` exhibits significant correlation, while the independently trained `pythia-6.9b` exhibits near zero correlation.

Suppose Alice trains an open-weight language model and Bob produces text using a derivative of Alice's model. Can Alice prove that Bob is using her model? We ground this question with a few motivating examples. Alice may be a model developer who suspects Bob of hosting a chat interface that violates Alice's terms of service.[3] Can Alice prove via querying Bob's interface that it is built (e.g., fine-tuned) from her model? Alternatively, Alice may suspect Bob of operating a social media bot using her model. Can she prove from the account's post history that it is using her model?

We formulate the question as an independence testing problem: the goal is to test whether Bob's text (or equivalently, the model producing it) is statistically independent of Alice's training run. The inherently randomized nature of language model training—due in part to the shuffling of training examples—allows us to treat the training outcome as a random variable. We assume access to the ordered sequence of training examples from Alice's training run and consider two variations of the problem depending on the type of access permitted to Bob's model. In the *query setting*, we are able to directly prompt Bob's model; in the *observational setting*, we merely observe text generated by Bob's model from an unknown prompt (or multiple prompts). We aim to design tests that are

1. **effective**—the test should have high power;
2. **transparent**—it should not rely on keeping model training or implementation details private;
3. **noninvasive**—it should not require Alice to modify her original training data or model.

In the query setting, prior work fails to achieve these three desiderata, compromising either transparency or noninvasiveness in order to retain effectiveness. For example, inserting canaries into Alice's model (e.g., by having it memorize a random binary string) [3] enables effective testing (based on whether Bob's model has memorized the canary) but requires modifying Alice's model. Using a small held-out test set [4] (and testing if Bob performs worse on the test data than Alice's training data) is only effective if the test set is private (since otherwise Bob can evade detection by training on the test set). In the observational setting, we are not aware of any effective tests (let alone those achieving all three desiderata). We defer a more thorough discussion of related work to Section 2.

The main insight underpinning the design of our tests is that language models tend to memorize their training data, and these memorization effects are stronger for more recent training examples. In this sense, language models exhibit *palimpsestic memorization*: all training examples influence the final trained model, but later examples diminish the effects of previous examples. Thus, we can test whether Bob's model or text derives from Alice's by designing test statistics that correlate its behavior with the ordering of examples in Alice's training run (Figure 1). In the query setting, we can directly evaluate the likelihood Bob's model gives to Alice's training examples (by evaluating his model on these examples to obtain token probabilities) and measure the correlation between example

---

[3]For example, Meta's terms of service require those using their models within a product to acknowledge that the product is "Built with Meta Llama 3" [2].

likelihood and training order. In the observational setting, we use various notions of n-gram overlap between a training example and Bob's text as proxies for its likelihood under Bob's model; we also try retraining copies of Alice's model on reshuffled data (from an intermediate checkpoint) to determine whether Bob's text has higher likelihood under Alice's original model versus the retrained copies. In both settings, if Alice has randomly shuffled her training data—as is common practice[4]—and Bob's model is independent of Alice's training run, then there is guaranteed to be no (statistically significant) correlation between likelihood (i.e., either the likelihood of Alice's training examples under Bob's model or the likelihood of Bob's text under Alice's model versus the retrained copies) and training order. Thus, our tests yield provable control over false positive errors.

We formalize our problem formulation and the implementation of our tests in Section 3. We empirically validate our tests in Section 4 using the Pythia (trained on `pythia` and `pythia-deduped`, the deduped and non-deduped Pile datasets used to train Pythia models) and OLMo (trained on `OLMo`, `OLMo-1.7`, and `OLMo-2`, the Dolma and OLMo-Mix datasets) model families, as well as small-scale models we train on TinyStories [5, 6, 7]. Finally, we conclude with a discussion of key takeaways and directions for future work in Section 5. We release code and data for reproducing experiments.[5]

## 2 Related Work

Our work develops methods for establishing the provenance of two types of artifacts that may derive from a language model: the model itself (in the query setting) and text (in the observational setting). We presently focus on the work most relevant to ours; see Oliynyk et al. [8], Wu et al. [9], Jawahar et al. [10], and Suvra et al. [11] for a more comprehensive surveys on model and text provenance. We also discuss prior work studying properties of memorization in language models.

**Model provenance.** Maini et al. [4] propose a method they term *dataset inference* that enables testing whether Bob's model is independent of Alice by measuring the difference in performance of Bob's model on Alice's training set versus a small i.i.d. held-out test set. Crucially, their method is only effective if the test set is kept private, thus precluding transparency. Though in principle it may be possible to achieve transparency by scaling up the size of the test set (e.g., to be commensurate with Alice's training set), doing so would necessarily be more invasive: it would not only require Alice to plan in advance for this specific scenario when launching her training run but moreover would also limit the amount of available training data. Finally, their method does not apply to the observational setting since it requires querying Bob's model on the held-out test set.

A line of work under the umbrella of *model fingerprinting* has developed a multitude of ways to perturb a language model in some subtle way (e.g., by planting a backdoor trigger [12, 13] or by embedding a pattern into the model's output [14, 15, 16]) so as to enable downstream identification of derivative models. Due to the perturbation step, these methods all fail to satisfy our noninvasive criterion. Model fingerprinting is also inapplicable to the observational setting since it requires querying Bob's model on the fingerprint trigger.

Another line of work on model provenance develops heuristic techniques for predicting whether two models are independent or not based on the similarity of their outputs [17, 18]; however, these methods are not always reliable and thus fail to satisfy our effectivity criterion. In particular, we show in Appendix C that independently trained models trained on similar data can behave more similarly to each other than actual derivative models. Finally, we remark that recent work developing independence tests for language models based on their weights is inapplicable to our setting [19, 20], since we do not assume access to the weights of Bob's model.

**Text provenance.** The problem of detecting whether a text derives from a language model has drawn considerable attention in the literature. While there exist a number of heuristic techniques for predicting whether text derives from a language model [21, 22], these heuristic techniques are inexact and often unreliable [10]; furthermore, often the focus is on determining whether a text derives from any language model (i.e., versus a human) rather than attributing text to a particular

---

[4]Language models are typically trained on randomly shuffled data; in particular, training consists of a sequence of gradient steps taken on batches of data sampled randomly from some large pool. While it is common to have distinct phases of training (e.g., pretraining, mid-training, and post-training), so long as data is still shuffled within a particular phase (e.g, pretraining) our tests will still be applicable (by considering the ordering specifically for that phase).

[5]`https://github.com/RohithKuditipudi/blackbox-model-tracing`.

language model. In contrast to these heuristic methods, our techniques provide provable control over false positives via exact p-values and enable attribution of text to (derivatives of) a particular language model. Finally, we remark that inference-time watermarks [23, 24, 25, 26]—i.e., techniques for detecting text downstream by applying watermarks to generated text—are inapplicable to our setting since we allow Bob (an untrusted party) to directly sample text himself. Other work has also shown that fine-tuning often destroys watermarks from the base model [27].

**Characterizing memorization in language models.** Our work crucially leverages properties of memorization in language models inspired by previous work, in particular that language models tend to memorize sequences seen in training and assign high likelihood to these sequences [28, 29]. However, with large-scale pre-training, models also forget examples seen earlier in training [30, 31]. Recent work studies the combined effects of memorization and forgetting by tracking which sequences occurred at a given step across training [31, 32, 33], where the sequence likelihood typically increases sharply after the first exposure and gradually decreases towards the mean. Building on top of these observations, we show that LLMs not only memorize more data seen later in training, but also, to an extent that is detectable, retain a finer-grained pre-training data order. Of particular relevance to our observational setting is that language models tend to regurgitate memorized phrases (from training data) even in contexts different from the ones seen in training [34, 35, 36, 37, 38], making traces of training data and order detectable from samples as well.

## 3   Methods

### 3.1   Problem formulation and testing framework

Let $\mathcal{X}$ be the vocabulary. We define a language model $\mu \in \Delta(\mathcal{X}^*)$ as a distribution over strings of text. We abstract a training algorithm $A \in \Delta(\mathcal{T})$ as a distribution over *transcripts* $\alpha \in \mathcal{T}$, which capture the full execution trace of a training run. We adopt the following setup:

1. Alice runs a training algorithm that produces a transcript $\alpha$, i.e., $\alpha \sim A$;
2. Bob produces an artifact $\beta$; and
3. Alice tests whether $\alpha \perp \beta$.

The type of artifact determines the problem setting. In the *query setting*, Bob produces a language model $\beta \in \Delta(\mathcal{X}^*)$; in the *observational setting*, Bob merely produces text $\beta \in \mathcal{X}^*$. Note that the observational setting is harder than the query setting since it provides strictly less information to Alice. In particular, if Alice has Bob's model then she can always generate text (given any prompt of her choosing) from the model herself. We will disambiguate the two cases by using $\mu_\beta \in \Delta(\mathcal{X}^*)$ to denote Bob's model and $x^\beta \in \mathcal{X}^*$ to denote Bob's text.

Throughout the remainder of the paper, we equate a transcript with a set of strings indexed by their training order, i.e., $\alpha = \{(x^i, t_i)\}_{i=1}^n \in (\mathcal{X}^* \times [N])^n$ for $n, N \in \mathbb{N}$, since this is the only part of the training outcome relevant to our tests. We assume that Alice trains on randomly shuffled data, producing a shuffled transcript (Assumption A1).

**Assumption A1.** A transcript $\alpha = \{(x^i, t_i)\}_{i=1}^n \in (\mathcal{X}^* \times [N])^n$ is *shuffled* if for any permutation $\sigma : [N] \to [N]$ the random variables $\{(x^i, t_i)\}_{i=1}^n$ and $\{(x^i, \sigma(t_i))\}_{i=1}^n$ are identically distributed.

Algorithm 1 abstracts a framework for obtaining exact p-values from arbitrary test statistics under Assumption A1, thus enabling provably exact control over false positive errors (Theorem 1). We will give concrete instantiations of various choices of test statistics for both the query and observational settings in Sections 3.2 and 3.3 respectively. A good test statistic should produce low p-values when Bob's artifact is not independent of Alice's transcript; in particular, we would like $\phi(\alpha, \beta)$ to be abnormally large when $\alpha$ and $\beta$ are not independent. Typically, we do not actually run Algorithm 1 in practice; rather, we efficiently simulate it (either exactly or approximately, depending on the test statistic) for large $m$ to enable obtaining small p-values at no additional computational cost.

**Theorem 1.** *Let $A$ satisfy Assumption A1. Let $\alpha \sim A$ and $\beta \perp \alpha$. Then the output of Algorithm 1 is uniformly distributed over $\{(j+1)/(m+1)\}_{j=0}^m$.*

*Proof.* From our assumption on $A$, it follows that the collection $\{\alpha_j\}_{j=1}^m$ comprises $m$ exchangeable copies of $\alpha$. The independence of $\alpha$ and $\beta$ thus implies $\{(\alpha_j, \beta)\}_{j=1}^m$ comprises $m$ exchangeable

**Algorithm 1:** Obtaining p-values from arbitrary test statistics

---

    **Input:** Transcript $\alpha = \{(x^i, t_i)\}_{i=1}^n$; artifact $\beta$
    **Parameters:** test statistic $\phi$; number of permutations $m$
    **Output:** p-value $\hat{p} \in (0, 1]$

1   **for** $j \in 1, \dots, m$ **do**
2      $\sigma_j \sim \text{Unif}([N] \to [N])$; $\alpha_j = \{(x^i, \sigma_j(t_i))\}_{i=1}^N$
3      $\phi_j \leftarrow \phi(\alpha_j, \beta)$
4   $\hat{p} \leftarrow 1 - \frac{1}{m+1}(1 + \sum_{j=1}^m \mathbf{1}\{\phi_j < \phi(\alpha, \beta)\})$ // `break ties randomly`
5   **return** $\hat{p}$

---

copies of $(\alpha, \beta)$. Because we break ties randomly, by symmetry it follows that $\phi(\alpha, \beta)$ will have uniform rank among $\{\phi_j\}_{j=1}^m$ and thus $\hat{p}$ is uniformly distributed over $\{\frac{j+1}{m+1}\}_{j=0}^m$.     □

In practice, we generally do not use the full transcript, which may comprise billions of training examples; instead, to save time we typically randomly subsample elements of the transcript (without replacement) and run our tests using the subsampled transcript. So long as the subsampled transcript satisfies Definition A1 (which will be the case if Alice shuffled her data and we take a random subsample), our tests remain valid.

### 3.2   Query setting

In the query setting, we assume Alice can directly obtain log-likelihoods from Bob's model $\mu_\beta$ for each training example. (Recall Bob's artifact $\beta = \mu_\beta$ in the query setting.) She can then correlate these log-likelihoods with the ordering of examples in the transcript $\alpha = \{(x^i, t_i)\}_{i=1}^n$ to test for independence. In particular, with $\rho$ denoting the Spearman rank correlation coefficient [39], let

$$\phi_{\text{query}}(\alpha, \mu_\beta) := \rho(\{\log \mu_\beta(x^i)\}_{i=1}^n, \{t_i\}_{i=1}^n) \tag{1}$$

We expect later-seen training examples to have higher likelihood according to Bob's model if it derives from Alice's training run, so we expect the statistic $\phi_{\text{query}}$ to be positive if Bob's model is not independent and close to zero if it is independent of Alice's training run.

The null distribution of the Spearman correlation coefficient is a known quantity—in particular, if $\alpha \perp \mu_\beta$ then $\phi(\alpha, \mu_\beta)$ from equation (1) follows a t-distribution with $n - 2$ degrees of freedom. Thus, in our experiments instead of explicitly running Algorithm 1, we instead directly convert $\phi$ to a p-value using the closed form cumulative distribution function for the t-distribution. This enables us to obtain extremely low, exact p-values without being computationally bottlenecked.

We can increase the power of our test by controlling for natural variation in text likelihoods (some texts are inherently less predictable than others) using an independent reference model $\mu_0$. In particular, let

$$\phi_{\text{query}}^{\text{ref}}(\alpha, \mu_\beta) := \rho\left(\left\{\log\left(\mu_\beta(x^i)/\mu_0(x^i)\right)\right\}_{i=1}^n, \{t_i\}_{i=1}^n\right) \tag{2}$$

The idea is to reduce the variance of the test statistic by subtracting the log-likelihood of the independent reference model. We can also regress the log-likelihood of Bob's model onto the reference model (and other features of text that may correlate with log-likelihood but are independent of Alice's training order) to better control for natural variation in text likelihood.

Finally, we can extend all of the test statistics in this section to the case where Alice only receives token predictions from Bob (instead of token probabilities) by first estimating token probabilities from Bob's predictions and then using these probabilities to apply the test statistics.

### 3.3   Observational setting

In the observational setting, we make no assumptions on how Bob generated the text $x^\beta$ that Alice observes. (Recall Bob's artifact $\beta = x_\beta$ in the observational setting.) Even if Bob did generate the text using a language model, we do not have access to the model and thus cannot compute its likelihood on her training examples as in the query setting. Instead, what we can do is compute the likelihood of Bob's text (or proxies thereof) under a collection of language models that we construct

to elicit correlations between Bob's text and Alice's example ordering. We take two approaches to constructing the collection by training models on either partitions or shuffles of Alice's transcript.

For the first approach (Algorithm 2, or $\phi_{\text{obs}}^{\text{part}}$), we train language models on different contiguous partitions of Alice's original ordered training data and correlate (some measure of) the likelihood of Bob's text under these models with their relative ordering. Motivated by previous work observing that language models sometimes regurgitate text verbatim from their training data [28, 29] and in particular are more likely to regurgitate text seen more recently in training [31, 32, 33], we use n-gram models and take $\chi$ to be either the likelihood of Bob's text under the model or the number of exact n-gram matches among Bob's text with the n-gram index underlying each model.

---

**Algorithm 2:** Training models on partitioned transcript ($\phi_{\text{obs}}^{\text{part}}$)

    **Input:** Transcript $\alpha = \{(x^i, t_i)\}_{i=1}^n$; text $x^\beta \in \mathcal{X}^*$
    **Parameters:** number of models $k$; metric $\chi$
    **Output:** test statistic $\phi_{\text{obs}}^{\text{part}}(\Gamma, \beta)$
**1** Sort examples $x$ by indices $t$
**2** Split sorted $x$ into $x_1, ..., x_k$ contiguous partitions and train models $\mu_1, ..., \mu_k$ on partitions
**3** **return** $\rho(\{\chi(\mu_j, x^\beta), \{j\}_{j=1}^k)$

---

For the second approach (Algorithm 3, or $\phi_{\text{obs}}^{\text{shuff}}$), we train neural language models on different shuffles of Alice's original ordered training data to determine whether Bob's text has abnormally high likelihood under the original ordering. In practice, we use the same model architecture as Alice (so that $\mu_0$ is Alice's model itself); we also repeat only a small fraction of her training run (e.g., the last 1-10%) to reduce computational costs. In our experiments, we take $\chi(\mu_i, x^\beta)$ to either be the likelihood of Bob's text $x^\beta$ under the model $\mu_i$ itself or the likelihood after finetuning each $\mu_i$ on Bob's text, with the motivation of the latter approach being to improve the robustness of our test to Bob finetuning or otherwise modifying Alice's model.

---

**Algorithm 3:** Training models on shuffled transcript ($\phi_{\text{obs}}^{\text{shuff}}$)

    **Input:** Transcript $\alpha = \{(x^i, t_i)\}_{i=1}^n$; text $x^\beta \in \mathcal{X}^*$
    **Parameters:** number of models $k$; metric $\chi$
    **Output:** test statistic $\phi_{\text{obs}}^{\text{shuff}}(\Gamma, \beta)$
**1** Sort examples $x$ by indices $t$
**2** Train model $\mu_0$ on $x$ (in sorted order) and models $\mu_1, ..., \mu_k$ on independent reshuffles of $x$
**3** $\mu \leftarrow (1/k) \sum_{i=1}^k \chi(\mu_i, x^\beta)$; $\sigma \leftarrow \sqrt{(1/(k-1)) \sum_{i=1}^k (\chi(\mu_i, x^\beta) - \mu)^2}$
**4** **return** $\left( \chi(\mu_0, x^\beta) - \mu \right) / \sigma$

---

Unlike the previous statistics, $\phi_{\text{obs}}^{\text{shuff}}$ does not apparently have a closed form output distribution under the null hypothesis. Obtaining exact p-values from $\phi_{\text{obs}}^{\text{shuff}}$ is possible via Algorithm 1 but would require retraining many models. Instead, in our experiments we report approximate p-values by treating the output of $\phi_{\text{obs}}^{\text{shuff}}$ as a z-score, and as a sanity check we report the degree to which these p-values empirically deviate from exact p-values under the null.

## 4 Experiments

### 4.1 Setup

**Transcript ($\alpha$).** We use the ordered pretraining data from various open-source language models as Alice's transcript. We consider five families of models, each corresponding to a different pretraining dataset ranging from 300B to 4T tokens: (1) `pythia`: The Pile dataset used for training Pythia models [5]; (2) `pythia-deduped`: The deduped version of The Pile used for training Pythia-deduped models; (3) `OLMo`: Dolma v1.5 dataset used for training OLMo models [6]; (4) `OLMo-1.7`: Dolma v1.7 dataset used for training OLMo-0424 and OLMo-0724 models; and (5) `OLMo-2`: OLMo-Mix used for training the stage1 of OLMo-2-1124 models [7]. Additionally, we use TinyStories [40] to train small-scale models for ablations that would otherwise be prohibitively expensive, such as trying multiple epochs of training. We subsample sequences from each dataset to conduct our test (see Appendices A and B regarding sampling details for the query and observational settings respectively).

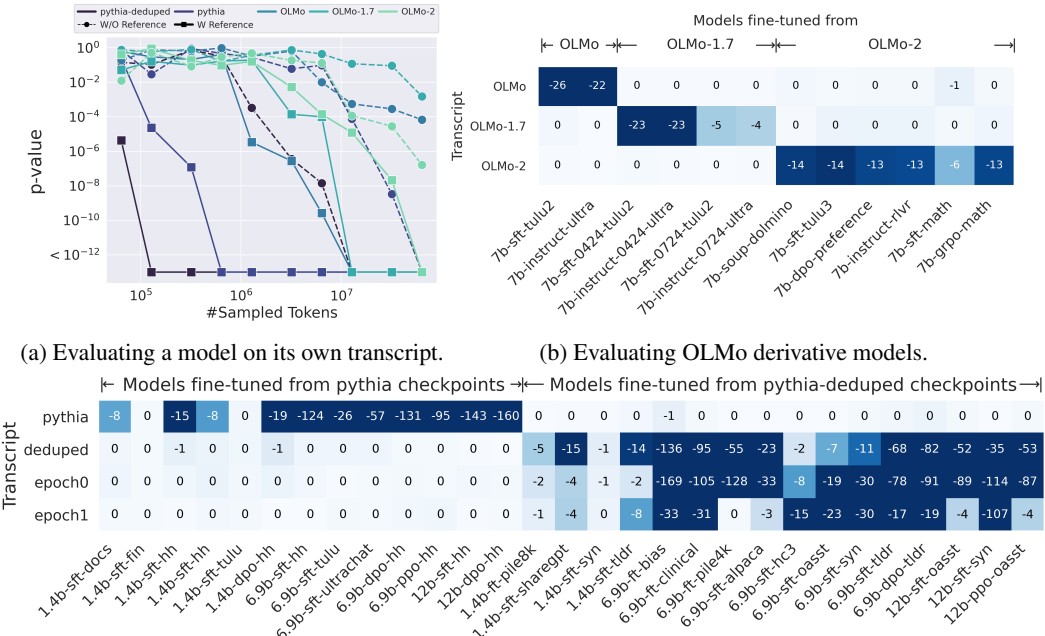

(a) Evaluating a model on its own transcript.          (b) Evaluating OLMo derivative models.

(c) Evaluating Pythia derivative models. For `pythia-deduped`, we try restricting queries to each epoch.

Figure 2: We report results (all p-values in $\log_{10}$) from $\phi_{\text{query}}^{\text{ref}}$ for 40 Pythia and OLMo derivative models in the query setting. We label each derivative model as {`model size`}-{`post-training method`}-{`post-training dataset`}.

**Artifact ($\beta$).** For Bob's artifact $\beta$, we use models that derive from one of the five families above or from our own TinyStories models. We vary the following factors when selecting the derivative models: (1) number of tokens on which the derived model has been finetuned, and (2) the type of finetuning, including supervised fine-tuning, preference optimization, and model souping.

## 4.2 Query setting

**Test statistic ($\phi$).** We report p-values using $\phi_{\text{query}}$ and $\phi_{\text{query}}^{\text{ref}}$ as we describe in Section 3.2.

**Reference model ($\mu_0$).** To implement $\phi_{\text{query}}^{\text{ref}}$, we consider three types of reference models $\mu_0$: (1) models pre-trained on an (almost) identical training dataset as Alice's, but using a different order, such as using Pythia models trained on the non-deduped version of Pile as references for Pythia models trained on the deduped version; (2) models pre-trained on other training datasets, e.g., using OLMo models as references for Pythia models; and (3) an ensemble of models from (1) and (2). In particular, $\mu_0$ must be known to be independent from $\Gamma$, in order to correctly test the independenence of $\mu$ and $\Gamma$. We present ablations on the type of reference model in Appendix A.5, and we find type (1) reference models yield the lowest p-values.

**Results.** Figure 2a displays the p-values we obtain over the number of tokens we query from the model (i.e., the size of the subsampled transcript multiplied by the number of tokens per example) when testing a model on its own training data (supposing Bob is using Alice's model without modification). For each of the five families, we use up to 1M 64-token sequences randomly sampled from the first epoch as our transcript $\alpha$ and evaluate the 7B-scale model checkpoint at the end of the first epoch. Crucially, though the actual correlation is typically small (between $0.001$ and $0.1$), the accuracy of the test increases with the number of queries; in other words, the effect size is small but statistically significant. Both $\phi_{\text{query}}$ and $\phi_{\text{query}}^{\text{ref}}$ obtain extremely small p-values with enough queries, but using a reference model allows us to reduce the query amount by an order of magnitude while still obtaining comparable p-values. Incidentally, our results strongly suggest that `pythia-2.8b-deduped`,[6] a model commonly used by the research community to study training

---

[6]https://huggingface.co/EleutherAI/pythia-2.8b-deduped

dynamics, was actually trained on the non-deduped version of Pile (with a p-value of $10^{-60}$) rather than the deduped version, contrary to its documentation.

Based on these results, we fix the number of tokens we query to be 100k for `pythia` and 5M for `OLMo`, `OLMo-1.7`, and `OLMo-2`. In Figures 2b and 2c, we respectively test 12 HuggingFace models that are fine-tuned from an OLMo checkpoint and 29 models that are fine-tuned from a Pythia checkpoint. These models cover common post-training techniques including supervised finetuning, preference optimization, and model souping (i.e., averaging multiple finetuned model weights); see Appendix A.2 for details. For Pythia finetunes, whose base models were trained for between one and two epochs (with independently shuffled data for each epoch), we test using random subsamples of the full transcript as well as each epoch's transcript. For all but four of these finetunes, we obtain p-values of at most $10^{-8}$ with at least one or both epochs (we can straighforwardly correct for multiple-testing by multiplying the p-values by 3). Notably, the four exceptions are all models at the smallest scale we test (1.4B), and the three models with p-values larger than $10^{-5}$ all incur significantly higher pretraining loss (ranging from 4.5 to 5.6) relative to their underlying base model ($3 \pm 0.1$); in these cases, Bob evades detection at the cost of significantly degrading the quality of his model relative to Alice. For OLMo finetunes, we obtain p-values less than $10^{-4}$ in every case and less than $10^{-13}$ in all but three cases. In all cases, we expect (based on Figure 2a) to be able to obtain even lower p-values with more queries.

Figure 2c already shows we are able to obtain low p-values for various Pythia derivatives by correlating their log-likelihood with the transcript of the first training epoch despite the fact that these models underwent a second (partial) epoch of training on reshuffled data. To further evaluate the effectiveness of our test on models trained for multiple epochs, we conduct a similar test with the `OLMo` and `OLMo-1.7` 7B models by using the first epoch's transcript to test the final checkpoints, which have respectively continued pretraining from the first epoch for 544B and 2300B tokens. We obtain a p-value below $10^{-10}$ using 64M query tokens for `OLMo` and below $10^{-4}$ using 256M tokens for `OLMo-1.7`. Furthermore, to stress test the effectiveness of our methods on models trained for many epochs, in Appendix A.4 we try training our own models on the Tinystories dataset for up to 10 epochs (reshuffling the data for each epoch). We observe that the final models' log-likelihood exhibits statistically significant correlation with the transcripts of many (though not all) previous epochs; like a palimpsest, the ordering of previous epochs are inscribed into the model along with later ones.

Finally, we conduct additional experiments to simulate the case where Alice only receives next token predictions from Bob instead of token probabilities (e.g., supposing Bob maintains an API that returns text responses to arbitrary prompts) in Appendix A.6. We find we are able to obtain low p-values even when using just a single query to (roughly) estimate token probabilities, thus incurring minimal overhead in this more challenging setting. We perform a cost analysis of running our tests in Appendix D for various existing model APIs.

## 4.3 Observational Setting

Recall we consider two approaches in the observational setting based on partitioning ($\phi_{\text{obs}}^{\text{part}}$) or reshuffling ($\phi_{\text{obs}}^{\text{shuff}}$) Alice's original training data, training language models on these data and evaluating these models on Bob's text.

### 4.3.1 Partitioning the transcript

**Test statistic ($\phi$).** We report p-values using $\phi_{\text{obs}}^{\text{part}}$ (Algorithm 2) as we describe in Section 3.3. We primarily experiment with a version of $\phi_{\text{obs}}^{\text{part}}$ using n-gram models ($n = 8$) wherein we let $\chi$ count the number of exact matches among Bob's text with the n-gram index underlying each model and let $k$ be the total number of minibatches in Alice's training run; in other words, we count the number of matching n-grams among Bob's text per each minibatch of training examples and correlate these counts with the minibatch order. See Appendix B.1 for additional implementation details.

**Sampling text.** Because the models we experiment with have limited context windows or are otherwise incapable of generating long, coherent texts, to obtain Bob's text $x^\beta$ we independently generate short texts then group these texts together (i.e., we treat $x^\beta$ as a collection of distinct documents and use $|x^\beta|$ to denote the total number of tokens across all documents). We generate these texts as continuations of prefixes from The Pile [41] dataset and vary the sampling temperature. Each prefix has 16 tokens and each continuation is at most 128 tokens.

**Results.** We experiment with Pythia models, whose n-gram training index (the dictionary mapping n-grams to batches) we can build with reasonable disk space (1.4TB) and query efficiently with the infini-gram code base [42]. We index a subsample of the transcript for the first 100K training batches and run our test on text we sample from subsequent training checkpoints up to the full 143K batches (treating the later checkpoints as finetunes). Because the test is costly to run for large numbers of tokens, we do not evaluate the full set of Pythia derivative models from earlier.

| step | $|x^\beta| = 640\text{K}$ | 1.28M | 3.20M | 6.40M | 12.8M | 19.2M |
|---|---|---|---|---|---|---|
| 100K | $3.0 \times 10^{-1}$ | $2.5 \times 10^{-2}$ | $2.2 \times 10^{-2}$ | $2.0 \times 10^{-3}$ | $6.3 \times 10^{-4}$ | $2.6 \times 10^{-4}$ |
| | $(8.6 \times 10^{-2}, 5.4 \times 10^{-1})$ | $(1.2 \times 10^{-2}, 5.3 \times 10^{-2})$ | $(3.7 \times 10^{-3}, 1.1 \times 10^{-1})$ | $(1.1 \times 10^{-3}, 4.0 \times 10^{-3})$ | $(8.2 \times 10^{-5}, 1.3 \times 10^{-3})$ | $(1.4 \times 10^{-4}, 3.0 \times 10^{-4})$ |
| 110K | $3.7 \times 10^{-1}$ | $2.6 \times 10^{-1}$ | $1.3 \times 10^{-1}$ | $9.3 \times 10^{-2}$ | $6.9 \times 10^{-2}$ | $4.4 \times 10^{-2}$ |
| | $(1.7 \times 10^{-1}, 7.1 \times 10^{-1})$ | $(1.1 \times 10^{-1}, 6.3 \times 10^{-1})$ | $(6.7 \times 10^{-2}, 2.4 \times 10^{-1})$ | $(8.7 \times 10^{-2}, 2.8 \times 10^{-1})$ | $(3.9 \times 10^{-2}, 8.3 \times 10^{-2})$ | $(3.3 \times 10^{-2}, 6.4 \times 10^{-2})$ |
| 120K | $4.5 \times 10^{-1}$ | $3.0 \times 10^{-1}$ | $3.9 \times 10^{-1}$ | $8.4 \times 10^{-2}$ | $7.4 \times 10^{-2}$ | $8.7 \times 10^{-2}$ |
| | $(1.9 \times 10^{-1}, 5.9 \times 10^{-1})$ | $(1.4 \times 10^{-1}, 6.4 \times 10^{-1})$ | $(1.3 \times 10^{-1}, 5.1 \times 10^{-1})$ | $(2.8 \times 10^{-2}, 1.7 \times 10^{-1})$ | $(1.2 \times 10^{-2}, 1.2 \times 10^{-1})$ | $(3.5 \times 10^{-2}, 1.4 \times 10^{-1})$ |
| 130K | $1.7 \times 10^{-1}$ | $5.1 \times 10^{-1}$ | $4.7 \times 10^{-1}$ | $1.7 \times 10^{-1}$ | $1.0 \times 10^{-1}$ | $4.4 \times 10^{-2}$ |
| | $(3.3 \times 10^{-2}, 4.5 \times 10^{-1})$ | $(4.6 \times 10^{-1}, 6.2 \times 10^{-1})$ | $(1.7 \times 10^{-1}, 8.5 \times 10^{-1})$ | $(1.0 \times 10^{-1}, 4.2 \times 10^{-1})$ | $(4.8 \times 10^{-2}, 1.5 \times 10^{-1})$ | $(4.1 \times 10^{-2}, 1.1 \times 10^{-1})$ |
| 143K | $5.9 \times 10^{-1}$ | $5.6 \times 10^{-1}$ | $3.9 \times 10^{-1}$ | $4.2 \times 10^{-1}$ | $4.3 \times 10^{-1}$ | $2.9 \times 10^{-1}$ |
| | $(2.9 \times 10^{-1}, 7.8 \times 10^{-1})$ | $(3.3 \times 10^{-1}, 7.0 \times 10^{-1})$ | $(3.2 \times 10^{-1}, 6.7 \times 10^{-1})$ | $(2.4 \times 10^{-1}, 6.0 \times 10^{-1})$ | $(2.8 \times 10^{-1}, 6.6 \times 10^{-1})$ | $(2.0 \times 10^{-1}, 5.9 \times 10^{-1})$ |

Table 1: We report median p-values and interquartile ranges over 10 trials from applying $\phi_{\text{obs}}^{\text{part}}$ to text sampled from `pythia-6.9b-deduped` checkpoints. Because sampling is expensive, for each trial we resample generations with replacement from an initial sample of 800K generations (i.e., 25.6M tokens), so the reported ranges are likely narrow.

In Table 1, even when testing the checkpoint corresponding to the end of the transcript (i.e., supposing Bob uses Alice's model directly) we require several hundred thousand tokens before we begin to observe consistently low p-values. These results suggest it is infeasible to use this test to detect when Bob is generating short snippets of text (e.g., social media posts) using Alice's model. However, we posit the test may be useful for larger-scale, ecosystem level analyses such as allowing Alice to estimate the total fraction of text on a social media platform that derives from her model. Moreover, the test is able to withstand a substantial amount of finetuning: with enough tokens, we observe decreasing p-values even after 30K steps (i.e., when Bob continues training Alice's model for up to 30% of its original pretraining budget).

We report additional results in Appendix B, varying the sampling parameters and token count. We find the p-values we obtain are somewhat sensitive to temperature. Notably, the power of our test diminishes substantially when generating text unconditionally with low temperature, which we attribute to low diversity among the generated texts.

### 4.3.2 Shuffling the transcript

**Test statistic ($\phi$).** We report p-values using $\phi_{\text{obs}}^{\text{shuff}}$ (Algorithm 2) as we describe in Section 3.3. We use the same model architecture as Alice and continue training from an intermediate checkpoint. We let $\chi$ be the likelihood of Bob's text under each model, either with or without finetuning the model on Bob's text. See Appendix B.2 for additional implementation details.

**Sampling text.** As before, to obtain Bob's text $x^\beta$ we independently generate short texts then group these texts together. We generate these texts as continuations of prefixes from the TinyStories test set. Each prefix has 20 tokens and each continuation is at most 32 tokens long.

**Results.** Running $\phi_{\text{obs}}^{\text{shuff}}$ requires retraining many models on reshuffled data, which is costly at scale. Thus, we conduct most of our experiments with small (3M) models we train ourselves on 500K documents from the Tinystories dataset. We vary three main factors: 1) the amount of data on which we retrain (ranging from the last 10-50K documents—i.e., the last 2-10% of training); 2) the amount of data (also sampled from the TinyStories dataset) on which Bob finetunes; and 3) the amount of text Bob generates. Unlike the tests in the previous section, the output of $\phi_{\text{obs}}^{\text{shuff}}$ is not an exact p-value but rather a z-score; nonetheless, for the sake of comparison to previous results, we report approximate p-values we obtain by converting z-scores to p-values under the assumption that the score distribution is normal. We defer the full details of our training and finetuning setup to Appendix B.2. We demonstrate the validity of our approximate p-values by empirically estimating their null distribution in Appendix B.5.

Even with retraining on as little as the last 2% of Alice's pretraining data, we are able to obtain p-values less than $10^{-3}$ from as few as 320 tokens in Bob's text (Figure 3, left panel). Moreover,

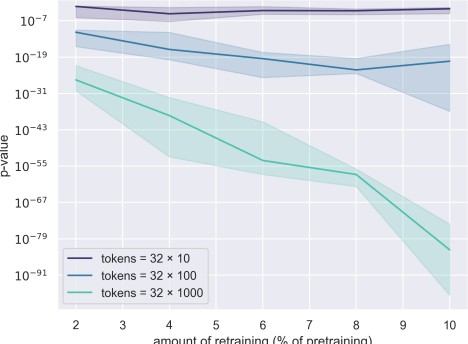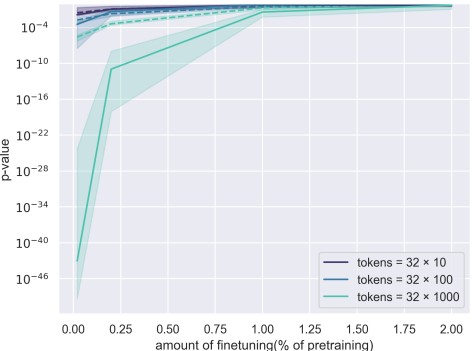

Figure 3: We report approximate p-values from applying $\phi_{\text{obs}}^{\text{shuff}}$ to TinyStories models, varying the number of tokens in Bob's text. (Left) We vary the amount of data we use to retrain the test models; we let $\chi$ be the log-likelihood of Bob's text under each model (no finetuning) and assume Bob uses Alice's model without modification. (Right) We fix the amount of retraining at 2% of Alice's pretraining data and vary the amount of data on which Bob finetunes Alice's model; we try $\phi_{\text{obs}}^{\text{shuff}}$ with (solid) and without (dashed) finetuning on Bob's text.

finetuning the retrained test models on Bob's text before evaluating their log-likelihood substantially improves the robustness of the test to Bob finetuning Alice's model (Figure 3, right panel); however, we require substantially more tokens in order to see a tangible benefit, and even with 32000 tokens both versions of the test lose power once Bob finetunes on as many documents as 1% of Alice's pretraining run. We include additional sweeps and ablations in Appendix B.4; notably, we find retraining the test models on more tokens improves the robustness of the test to finetuning.

To extend beyond toy experiments, we leverage existing OLMo 2 checkpoints—from the cooldown phase of their pretraining run—trained on different random shuffles of the same 50B tokens (representing roughly 1% of the pretraining data). In particular, there are three such checkpoints for both `OLMo-2-0425-1B` and `OLMo-2-1124-7B`. We use these checkpoints to simulate running $\phi_{\text{obs}}^{\text{shuff}}$ with $k = 2$ (and no finetuning on Bob's text) by generating text from one of the checkpoints (using the same setup and prompt distribution as in the previous experiment on TinyStories) and treating the other two checkpoints as test models. Given 640 tokens of Bob's text, we obtain median p-values (with interquartile ranges parenthesized) of $6.25 \times 10^{-3}$ $(6.44 \times 10^{-5}, 1.44 \times 10^{-1})$ at the 1B scale and $1.44 \times 10^{-3}$ $(1.01 \times 10^{-6}, 6.00 \times 10^{-2})$ at the 7B scale; using only two test models makes the p-values more volatile. Additional results are in Appendix B.6.

## 5 Discussion

We develop exact tests that enable language model providers and developers to determine: 1) whether another language model is trained independently of their model and 2) whether a text is generated independently of their model. Our tests are transparent and noninvasive by design, and we evaluate their effectiveness with a number of models across a range of scales and training recipes. We presently remark on some limitations and directions for future work.

Recall that we assume the role of Alice, so we have access to Alice's training data in order to carry out our tests. In practice, model developers (including developers of open-weight models) may be reluctant to disclose their training data due to concerns around copyright liability or competitive advantage (of course, any model developer can still use our tests internally). To enable independent third-party verification of test results under these circumstances, we propose running our tests using a subset of the transcript that the model developer is willing to disclose (e.g., the relative ordering of Wikipedia documents in the pretraining data).

All our tests are somewhat costly to run, either because they require many tokens from Bob to be effective or retraining a collection of language models. Bringing these costs down is an important direction for future work. Particularly in the observational setting, where Alice cannot simply spend more queries to obtain more tokens, designing more lightweight tests than $\phi_{\text{obs}}^{\text{shuff}}$ with similar token complexity is an important open problem.

Finally, beyond our specific problem setting, our experiments yield new insights into memorization in language models. Exploring the implications of these findings on issues of privacy and copyright and for designing more effective models an exciting direction for future work.

## Acknowledgments

We would like to thank Jiacheng Liu and Pete Walsh for their help on reconstructing the OLMo training data. This research is supported in part by grants from Google and Open Philanthropy.

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

# Technical Appendices and Supplementary Material

## A Query Setting Experimental Details and Additional Results

### A.1 Query distribution

When sampling sequences from Alice's training dataset to construct queries from her transcript $\alpha$, we vary the following hyperparameters: (1) the length of the sequence $L$, (2) the training step $s$ at which the sequence occurs (the last training step in case a sequence occurs multiple times in training), and (3) the start token position $t$ of the sequence in the original training example.

All three hyperparameters affect the computation of likelihood used in $\phi_{\text{query}}^{\text{ref}}$. Specifically, given a sequence of $L$ tokens $x_{s,t}, \ldots, x_{s,t+L}$, we compute the average language modeling loss $\ell$ (i.e., negative log-likelihood) across all tokens as:

$$\ell = \frac{1}{L-1} \sum_{t < i \leq t+L} -\log(\mu(x_{s,i}|x_{s,t} \ldots x_{s,i-1}))$$

We provide the experiment details below on `OLMo-7B`. Overall, we observe that for a fixed number of sampled tokens (i.e., the number of sampled sequences $\times$ the length of the sequence), sampling from the first few tokens of each training example, i.e., $t = 0$, and using sequences that occur closer to the testing checkpoint provide the strongest memorization signals. As we discussed in Section 5, these results also provide insights into LLM memorization behaviors, characterizing which sequences are more likely to be memorized by the model.

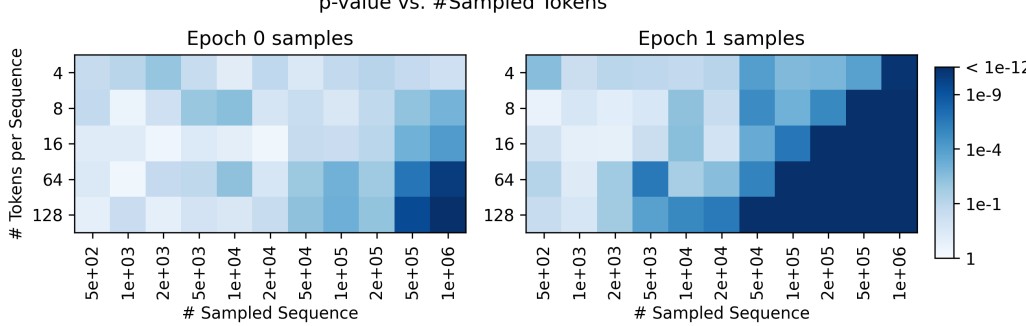

Figure 4: We vary the sequence length and number of sequences in the transcript $\Gamma$ in computing $\phi_{\text{query}}^{\text{ref}}$ for `OLMo-7B`. P-values are affected by the total number of tokens $\Gamma$ and the training step at which the sequence occurs, i.e., Epoch 1 sequences provide more signals than Epoch 0 sequences.

**Sequence length.** We measure how sequence length affects our test results using $\phi_{\text{query}}^{\text{ref}}$ in Figure 4. Given a fixed number of sequence samples, increasing the sequence length $L$ leads to a smaller p-value, i.e., for each column, p-value decreases as we move from top to bottom. However, given a fixed budget of tokens, sequence length has *no significant effect* on the p-value. We can see the diagonal trend of low p-values from the Epoch 1 samples (and slightly in the Epoch 0 samples). Hence, the total number of tokens in $\alpha$ is most relevant in the value of $\phi_{\text{query}}^{\text{ref}}$, rather than the number of sequences or sequence length individually.

**Training step.** Comparing Figure 4 Left and Right, we show that samples from Epoch 1 yield smaller p-values given the same sample size, suggesting sequences seen later in training are more likely to be memorized by the model.

**Start token position.** `OLMo-7B` uses a context window of 2048 tokens, so we can ablate the position of the tokens of sampled text. We sample 64-token sequences starting at position $t = 0/256/512/1024/1536/1920$. Figure 5 shows that sampling sequences at position 0 have the strongest memorization signals across all sampling sizes.

### A.2 Finetuning results

We provide the raw results for the fine-tuned models shown in Section 4.2, Figure 2b and 2c.

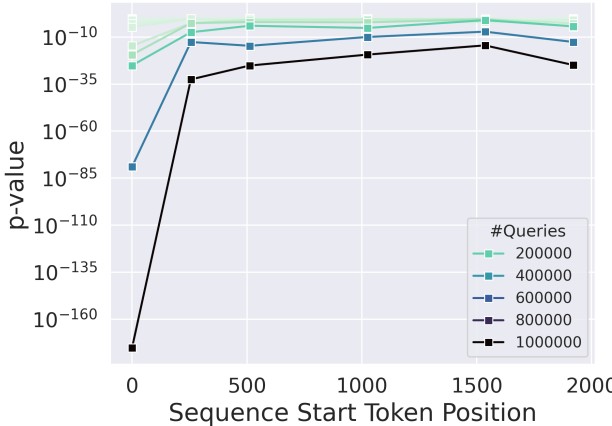

Figure 5: We vary the sample token position and number of queries in computing $\phi_{\text{query}}^{\text{ref}}$ for `OLMo-7B` models. Sampling from the start of an sequence consistently yields smaller p-value across all sample size.

We evaluate $\phi_{\text{query}}^{\text{ref}}$ on models from HuggingFace against the `pythia` and `pythia-deduped` models of different sizes. For each of the listed models, we use transcripts (training orders) of `pythia`, `pythia-deduped`, `pythia-deduped` Epoch 0, and `pythia-deduped` Epoch 1. In Table 2,3 we evaluate Pythia model derivatives and in Table 4 we evaluate non-derivatives, such as OLMo family models. In Table 5, we evaluate OLMo fine-tuned models using transcripts of `OLMo`, `OLMo-1.7`, `OLMo-2`. The transcript $\alpha$ consists of 100k randomly sampled sequences of length 64 for Pythia and 1M sequences for OLMo (except for `OLMo-1.7`, for which we use 5M sequences since the training order of the second epoch is not available. Instead we must use samples from the first epoch, which have weaker memorization effects).

| | p-value w.r.t. different training data order | | | |
| Model Name | pythia | pythia-deduped | Epoch 0 | Epoch 1 |
|---|---|---|---|---|
| **Base Models Trained on `pythia`** | | | | |
| `EleutherAI/pythia-1.4b` | $9.6 \times 10^{-24}$ | $1.5 \times 10^{-1}$ | $9.3 \times 10^{-1}$ | $3.3 \times 10^{-1}$ |
| `EleutherAI/pythia-2.8b` | $5.1 \times 10^{-62}$ | $3.5 \times 10^{-1}$ | $8.2 \times 10^{-1}$ | $7.8 \times 10^{-1}$ |
| `EleutherAI/pythia-6.9b` | $4.2 \times 10^{-142}$ | $-$ | $-$ | $-$ |
| `EleutherAI/pythia-6.9b-v0` | $2.0 \times 10^{-67}$ | $1.6 \times 10^{-2}$ | $1.3 \times 10^{-1}$ | $8.0 \times 10^{-2}$ |
| `EleutherAI/pythia-12b` | $1.3 \times 10^{-184}$ | $9.5 \times 10^{-1}$ | $4.1 \times 10^{-1}$ | $4.8 \times 10^{-1}$ |
| **Base Models Trained on `pythia-deduped`** | | | | |
| `EleutherAI/pythia-1.4b-deduped` | $9.6 \times 10^{-1}$ | $5.4 \times 10^{-23}$ | $9.3 \times 10^{-13}$ | $1.8 \times 10^{-14}$ |
| `EleutherAI/pythia-1.4b-deduped-v0` | $3.6 \times 10^{-1}$ | $1.0 \times 10^{-12}$ | $2.7 \times 10^{-10}$ | $4.4 \times 10^{-5}$ |
| `EleutherAI/pythia-2.8b-deduped` | $1.2 \times 10^{-60}$ | $9.1 \times 10^{-1}$ | $9.1 \times 10^{-1}$ | $8.1 \times 10^{-1}$ |
| `EleutherAI/pythia-2.8b-deduped-v0` | $7.3 \times 10^{-1}$ | $3.3 \times 10^{-79}$ | $7.1 \times 10^{-54}$ | $1.4 \times 10^{-34}$ |
| `EleutherAI/pythia-6.9b-deduped` | $-$ | $7.7 \times 10^{-135}$ | $1.8 \times 10^{-173}$ | $1.2 \times 10^{-38}$ |
| `EleutherAI/pythia-6.9b-deduped-v0` | $7.4 \times 10^{-1}$ | $9.3 \times 10^{-60}$ | $1.3 \times 10^{-236}$ | $5.7 \times 10^{-228}$ |
| `EleutherAI/pythia-12b-deduped` | $9.3 \times 10^{-1}$ | $7.6 \times 10^{-158}$ | $2.5 \times 10^{-233}$ | $4.3 \times 10^{-23}$ |
| `EleutherAI/pythia-12b-deduped-v0` | $4.5 \times 10^{-1}$ | $2.5 \times 10^{-86}$ | $0.0 \times 10^{0}$ | $3.7 \times 10^{-301}$ |

Table 2: We compute p-values using $\phi_{\text{query}}^{\text{ref}}$ for different Pythia base model variants [5].

The p-values of running $\phi_{\text{query}}^{\text{ref}}$ with the different transcripts are reported in Tables 3, 4, and 5. The blue boxes highlight p-values that are less than $10^{-2}$.

As we discussed in Section 4.2, `pythia-2.8b-deduped` has a non-significant p-value with the `pythia-deduped` training order, but a p-value of $1.2 \times 10^{-60}$ with the `pythia` transcript, i.e., the non-deduped data used to train `pythia-2.8b`, which suggests there is likely mislabeling of the HuggingFace model.

| Model Name | p-value w.r.t. different training data order | | | |
| | pythia | pythia-deduped | Epoch 0 | Epoch 1 |
|---|---|---|---|---|
| **Fine-tuned Models with Base Models Trained on pythia** | | | | |
| herMaster/pythia1.4B-finetuned-on-lamini-docs | $7.4 \times 10^{-9}$ | $3.2 \times 10^{-1}$ | $2.5 \times 10^{-1}$ | $5.9 \times 10^{-1}$ |
| LinguaCustodia/fin-pythia-1.4b | $6.4 \times 10^{-1}$ | $6.0 \times 10^{-1}$ | $6.1 \times 10^{-1}$ | $5.9 \times 10^{-1}$ |
| lomahony/pythia-1.4b-helpful-sft | $3.8 \times 10^{-16}$ | $5.6 \times 10^{-2}$ | $2.9 \times 10^{-1}$ | $8.4 \times 10^{-1}$ |
| Leogrin/eleuther-pythia1b-hh-sft | $1.9 \times 10^{-9}$ | $1.7 \times 10^{-1}$ | $1.0 \times 10^{-1}$ | $4.1 \times 10^{-1}$ |
| kykim0/pythia-1.4b-tulu-v2-mix | $3.3 \times 10^{-1}$ | $3.0 \times 10^{-1}$ | $4.3 \times 10^{-1}$ | $4.8 \times 10^{-1}$ |
| lomahony/pythia-1.4b-helpful-dpo | $9.4 \times 10^{-20}$ | $1.4 \times 10^{-2}$ | $1.2 \times 10^{-1}$ | $9.0 \times 10^{-1}$ |
| lomahony/eleuther-pythia6.9b-hh-sft | $4.9 \times 10^{-125}$ | $2.0 \times 10^{-1}$ | $4.2 \times 10^{-1}$ | $9.0 \times 10^{-1}$ |
| allenai/open-instruct-pythia-6.9b-tulu | $4.8 \times 10^{-27}$ | $2.5 \times 10^{-1}$ | $8.6 \times 10^{-1}$ | $5.4 \times 10^{-1}$ |
| pkarypis/pythia-ultrachat | $1.3 \times 10^{-58}$ | $2.4 \times 10^{-1}$ | $6.5 \times 10^{-1}$ | $5.9 \times 10^{-1}$ |
| lomahony/eleuther-pythia6.9b-hh-dpo | $1.5 \times 10^{-132}$ | $6.3 \times 10^{-1}$ | $4.9 \times 10^{-1}$ | $4.2 \times 10^{-1}$ |
| usvsnsp/pythia-6.9b-ppo | $6.4 \times 10^{-96}$ | $6.5 \times 10^{-1}$ | $6.6 \times 10^{-1}$ | $2.7 \times 10^{-1}$ |
| lomahony/eleuther-pythia12b-hh-sft | $4.9 \times 10^{-144}$ | $8.4 \times 10^{-1}$ | $4.4 \times 10^{-1}$ | $7.7 \times 10^{-1}$ |
| lomahony/eleuther-pythia12b-hh-dpo | $2.6 \times 10^{-161}$ | $9.0 \times 10^{-1}$ | $5.0 \times 10^{-1}$ | $6.0 \times 10^{-1}$ |
| **Fine-tuned Models with Base Models Trained on pythia-deduped** | | | | |
| naxautify/pythia-1.4b-deduped-8k | $9.4 \times 10^{-1}$ | $4.9 \times 10^{-6}$ | $2.9 \times 10^{-3}$ | $4.7 \times 10^{-2}$ |
| HWERI/pythia-1.4b-deduped-sharegpt | $8.3 \times 10^{-1}$ | $1.7 \times 10^{-16}$ | $2.2 \times 10^{-5}$ | $1.2 \times 10^{-5}$ |
| lambdalabs/pythia-1.4b-deduped-synthetic-instruct | $8.5 \times 10^{-1}$ | $2.7 \times 10^{-2}$ | $9.5 \times 10^{-2}$ | $1.5 \times 10^{-1}$ |
| trl-lib/pythia-1b-deduped-tldr-sft | $4.2 \times 10^{-1}$ | $3.4 \times 10^{-15}$ | $4.3 \times 10^{-3}$ | $4.0 \times 10^{-9}$ |
| lambdalabs/pythia-6.9b-deduped_alpaca | $9.4 \times 10^{-1}$ | $7.8 \times 10^{-24}$ | $2.5 \times 10^{-34}$ | $4.1 \times 10^{-4}$ |
| EleutherAI/pythia-intervention-6.9b-deduped | $6.4 \times 10^{-2}$ | $5.1 \times 10^{-137}$ | $4.0 \times 10^{-170}$ | $6.1 \times 10^{-34}$ |
| cc0de/EleutherAI-pythia-6.9b-deduped-full-ft-clinical-bsz16-lr5e-06 | $5.6 \times 10^{-1}$ | $3.0 \times 10^{-96}$ | $2.0 \times 10^{-106}$ | $3.3 \times 10^{-32}$ |
| pszemraj/pythia-6.9b-HC3 | $9.4 \times 10^{-1}$ | $2.3 \times 10^{-3}$ | $1.6 \times 10^{-9}$ | $6.1 \times 10^{-16}$ |
| dvruette/oasst-pythia-6.9b-4000-steps | $8.8 \times 10^{-1}$ | $2.4 \times 10^{-8}$ | $1.0 \times 10^{-20}$ | $1.4 \times 10^{-24}$ |
| CarperAI/pythia-6.9b-deduped-4k | $8.9 \times 10^{-1}$ | $9.0 \times 10^{-56}$ | $3.4 \times 10^{-129}$ | $5.8 \times 10^{-1}$ |
| lambdalabs/pythia-6.9b-deduped-synthetic-instruct | $5.3 \times 10^{-1}$ | $1.6 \times 10^{-12}$ | $2.1 \times 10^{-31}$ | $1.9 \times 10^{-31}$ |
| HuggingFaceH4/EleutherAI_pythia-6.9b-deduped__sft__tldr | $3.0 \times 10^{-1}$ | $1.2 \times 10^{-69}$ | $4.2 \times 10^{-79}$ | $3.6 \times 10^{-18}$ |
| trl-lib/pythia-6.9b-deduped-tldr-offline-dpo | $1.6 \times 10^{-1}$ | $1.1 \times 10^{-83}$ | $7.3 \times 10^{-92}$ | $2.6 \times 10^{-20}$ |
| OpenAssistant/pythia-12b-sft-v8-7k-steps | $5.7 \times 10^{-1}$ | $3.8 \times 10^{-53}$ | $1.4 \times 10^{-90}$ | $3.9 \times 10^{-5}$ |
| lambdalabs/pythia-12b-deduped-synthetic-instruct | $9.6 \times 10^{-1}$ | $5.9 \times 10^{-36}$ | $3.6 \times 10^{-115}$ | $2.4 \times 10^{-108}$ |
| OpenAssistant/pythia-12b-sft-v8-rlhf-2k-steps | $4.8 \times 10^{-1}$ | $7.1 \times 10^{-54}$ | $2.9 \times 10^{-88}$ | $2.0 \times 10^{-5}$ |

Table 3: We compute p-values using $\phi_{\text{query}}^{\text{ref}}$ for different Pythia fine-tuned model variants [5, 43, 44]. Fine-tuned models are listed in the same order as in Figure 2c from left to right.

| Model Name | P-Value w.r.t. Different Training Data Order | | | |
| | pythia | pythia-deduped | Epoch 0 | Epoch 1 |
|---|---|---|---|---|
| **Base Models Trained on Neither of the Two Orders** | | | | |
| EleutherAI/pythia-6.9b-deduped-v0-seed42 | $5.9 \times 10^{-1}$ | $5.4 \times 10^{-1}$ | $8.5 \times 10^{-1}$ | $4.9 \times 10^{-1}$ |
| allenai/OLMo-7B-hf | $7.9 \times 10^{-1}$ | $5.8 \times 10^{-1}$ | $6.4 \times 10^{-1}$ | $6.5 \times 10^{-1}$ |
| allenai/OLMo-7B-0424-hf | $7.6 \times 10^{-1}$ | $5.7 \times 10^{-1}$ | $5.9 \times 10^{-1}$ | $2.8 \times 10^{-1}$ |
| allenai/OLMo-7B-0724-hf | $8.8 \times 10^{-1}$ | $6.3 \times 10^{-1}$ | $6.6 \times 10^{-1}$ | $3.1 \times 10^{-1}$ |
| allenai/OLMo-2-1124-7B | $1.5 \times 10^{-1}$ | $2.9 \times 10^{-1}$ | $6.1 \times 10^{-1}$ | $2.3 \times 10^{-1}$ |
| meta-llama/Meta-Llama-3-8B | $2.3 \times 10^{-1}$ | $1.3 \times 10^{-1}$ | $7.8 \times 10^{-1}$ | $5.4 \times 10^{-1}$ |

Table 4: We compute p-values using $\phi_{\text{query}}^{\text{ref}}$ for models that are not trained on pythia or pythia-deduped. All p-values are above 0.1, indicting the null hypothesis is true, which matches the ground truth.

## A.3 Continued pretraining results

We provide additional analysis for continued pretraining in Section 4.2 on Pythia models. Specifically, we choose a base checkpoint $A$ of pythia-deduped (e.g., the step100000 checkpoint) and a finetune

| | P-Value w.r.t. Different Training Data Order | | |
| Model Name | `OLMo` | `OLMo-1.7` | `OLMo-2` |
|---|---|---|---|
| **Fine-tuned Models with Base Models Trained on `OLMo`** | | | |
| `allenai/OLMo-7B-SFT-hf` | $1.9 \times 10^{-27}$ | $3.8 \times 10^{-1}$ | $7.9 \times 10^{-1}$ |
| `allenai/OLMo-7B-Instruct-hf` | $1.3 \times 10^{-23}$ | $9.3 \times 10^{-1}$ | $9.6 \times 10^{-1}$ |
| **Fine-tuned Models with Base Models Trained on `OLMo-1.7`** | | | |
| `allenai/OLMo-7B-0424-SFT-hf` | $8.2 \times 10^{-1}$ | $7.7 \times 10^{-24}$ | $6.5 \times 10^{-1}$ |
| `allenai/OLMo-7B-0424-Instruct-hf` | $8.4 \times 10^{-1}$ | $4.0 \times 10^{-24}$ | $7.9 \times 10^{-1}$ |
| `allenai/OLMo-7B-0724-SFT-hf` | $7.6 \times 10^{-1}$ | $4.4 \times 10^{-6}$ | $9.6 \times 10^{-1}$ |
| `allenai/OLMo-7B-0724-Instruct-hf` | $8.4 \times 10^{-1}$ | $1.5 \times 10^{-5}$ | $7.8 \times 10^{-1}$ |
| **Fine-tuned Models with Base Models Trained on `OLMo-2`** | | | |
| `allenai/OLMo-2-1124-7B` | $4.2 \times 10^{-1}$ | $7.0 \times 10^{-1}$ | $5.4 \times 10^{-15}$ |
| `allenai/OLMo-2-1124-7B-SFT` | $6.3 \times 10^{-1}$ | $1.5 \times 10^{-1}$ | $1.9 \times 10^{-15}$ |
| `allenai/OLMo-2-1124-7B-DPO` | $4.7 \times 10^{-1}$ | $1.2 \times 10^{-1}$ | $1.8 \times 10^{-14}$ |
| `allenai/OLMo-2-1124-7B-Instruct` | $4.9 \times 10^{-1}$ | $1.3 \times 10^{-1}$ | $3.1 \times 10^{-14}$ |
| `Neelectric/OLMo-2-1124-7B-Instruct_SFT` | $7.1 \times 10^{-2}$ | $5.4 \times 10^{-1}$ | $1.7 \times 10^{-7}$ |
| `Neelectric/OLMo-2-1124-7B-Instruct_GRPO` | $4.8 \times 10^{-1}$ | $2.1 \times 10^{-1}$ | $2.2 \times 10^{-14}$ |

Table 5: We compute p-values using $\phi_{\text{query}}^{\text{ref}}$ for different OLMo model variants.

checkpoint $B$ after it (e.g., the step120000 checkpoint), and evaluate $B$ on $A$'s data and ordering (only samples seen before step100000). We compute $\phi_{\text{query}}^{\text{ref}}$ 40K samples with 64 tokens each for all valid base and checkpoint model pairs from step70000 to step140000, in Figure 6. We use `pythia` as the reference model. Even after the step70000 checkpoint was trained for an additional 70000 steps, the p-value between the step70000 and step140000 models is very small (2.23e-66). We can also see, as perhaps expected, that the p-values are much lower for the 6.9b-parameter models than the 1.4b, 1b, and 410m-parameter models—i.e., the larger model memorizes its training data more.

### A.4 Training for multiple epochs on TinyStories

We train a Transformer-architecture model ($\texttt{d\_model} = 256, \texttt{d\_ffn} = 512, \texttt{num\_layers} = 4$, approximately 3M parameters) on the TinyStories dataset [40] for 10 epochs of 50k samples. We run $\phi_{\text{query}}^{\text{ref}}$ with the Spearman correlation p-value on the training dataset between the model trained at Epoch $i$ for $i = 0, \ldots, 9$ (`model_epoch`) on the data ordering of Epoch $j$ for $j = 0, \ldots, 9$ (`order_epoch`). The results when using $N = 50$K and 20K samples are shown in Figure 7. Although variable, memorization effects are strong in general for at least 3 epochs, and the p-value is low in general for the recent 4 or 5 epochs. We use a reference model that uses the same architecture and training strategy but a different training order seed.

### A.5 Reference model ablations

For our experiments, we have used reference models very similar to the candidate model being audited—for example, the `pythia-6.9b` model as the reference model for testing `pythia-6.9b-deduped`. We ablate the reference model $\mu_0$ used in $\phi_{\text{query}}^{\text{ref}}$ and compare the p-values we get in Table 6. The averaged Llama models are:

- `meta-llama/Meta-Llama-3-8B`;
- `meta-llama/Llama-2-7b-hf`; and
- `huggyllama/llama-7b`,

and the averaged Pythia models are

- `EleutherAI/pythia-6.9b`;
- `EleutherAI/pythia-2.8b`;
- `EleutherAI/pythia-1.4b`;
- `EleutherAI/pythia-1b`;
- `EleutherAI/pythia-410m`;

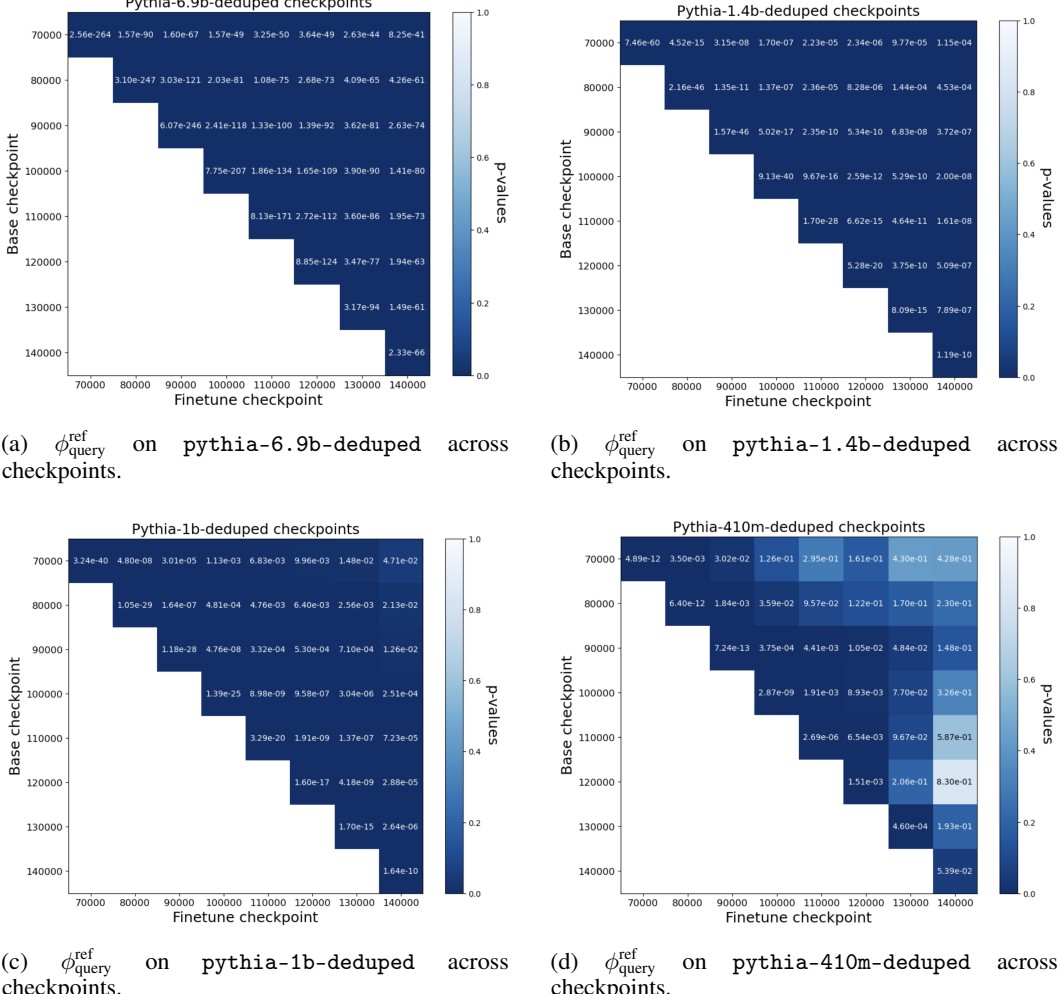

Figure 6: We run $\phi_{\text{query}}^{\text{ref}}$ using 40k samples on `pythia-6.9b-deduped`, `pythia-1.4b-deduped`, `pythia-1b-deduped`, and `pythia-410m-deduped` checkpoints. We find p-values are low even after significant continued pre-training (70k more steps after the 70k checkpoint), and also that memorization effects are smaller for smaller models.

- `EleutherAI/pythia-160m`;
- `EleutherAI/pythia-2.8b-deduped`;
- `EleutherAI/pythia-1.4b-deduped`;
- `EleutherAI/pythia-1b-deduped`;
- `EleutherAI/pythia-410m-deduped`; and
- `EleutherAI/pythia-160m-deduped`.

We can see that there is little signal without using a reference model to account for inherent variations in perplexity of text (the first row). For the later rows, based off the p-values, we conclude that the best reference models are models from the same family, and of similar capability. With 20M tokens, we get p-values significant at 1e-2 for all reference models (and combinations) tested.

### A.6   Estimating logprobs from generated text

Let's consider the case where Bob's API only provides access to the generated text without token probabilities. We show that Alice can still approximate logprobs by repeatedly querying Bob's model.

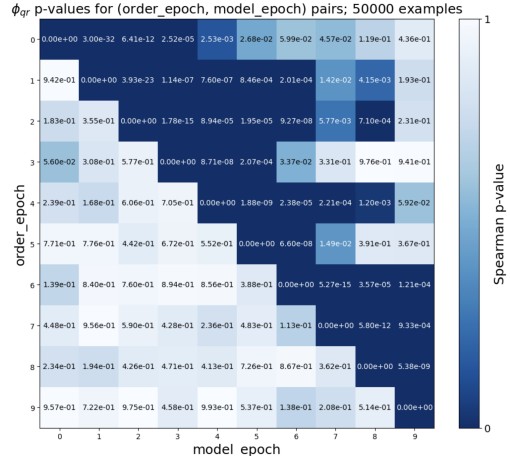
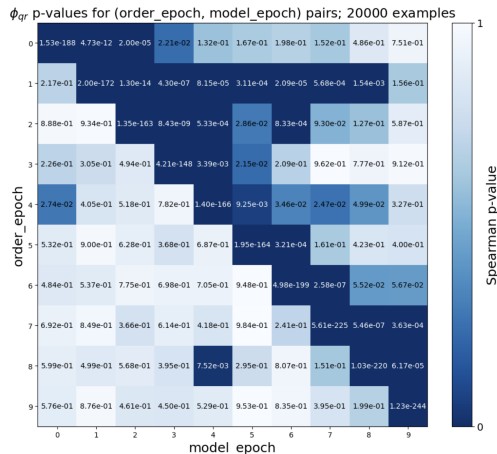

(a) $\phi_{\text{query}}^{\text{ref}}$ between order epoch and model epoch for 50k eval samples.

(b) $\phi_{\text{query}}^{\text{ref}}$ between order epoch and model epoch for 20k eval samples.

Figure 7: We train a Transformer model on TinyStories samples for 10 epochs (of 50k sequences), each of a random data ordering. We report p-values from the 10 epochs on the 10 data orders, which remain small even after 10 epochs.

| Reference Model | 20M Tokens | 12M Tokens | 4M Tokens |
|---|---|---|---|
| None | $2.1 \times 10^{-1}$ | $5.9 \times 10^{-1}$ | $8.6 \times 10^{-2}$ |
| EleutherAI/pythia-6.9b | $8.0 \times 10^{-20}$ | $3.1 \times 10^{-15}$ | $2.6 \times 10^{-5}$ |
| EleutherAI/pythia-2.8b | $1.0 \times 10^{-15}$ | $7.0 \times 10^{-12}$ | $3.7 \times 10^{-3}$ |
| EleutherAI/pythia-1.4b | $8.1 \times 10^{-12}$ | $3.4 \times 10^{-9}$ | $1.3 \times 10^{-2}$ |
| EleutherAI/pythia-1b | $7.3 \times 10^{-10}$ | $6.4 \times 10^{-7}$ | $5.5 \times 10^{-2}$ |
| EleutherAI/pythia-410m | $2.5 \times 10^{-6}$ | $2.0 \times 10^{-4}$ | $2.3 \times 10^{-1}$ |
| EleutherAI/pythia-160m | $1.2 \times 10^{-3}$ | $1.2 \times 10^{-2}$ | $9.0 \times 10^{-1}$ |
| Averaged Pythia models | $1.7 \times 10^{-7}$ | $1.3 \times 10^{-5}$ | $4.2 \times 10^{-4}$ |
| meta-llama/Meta-Llama-3-8B | $7.9 \times 10^{-3}$ | $9.2 \times 10^{-3}$ | $1.6 \times 10^{-1}$ |
| Averaged Llama models | $3.9 \times 10^{-3}$ | $1.2 \times 10^{-2}$ | $7.4 \times 10^{-1}$ |

Table 6: We compute p-values using different reference models $\mu_0$ with $\phi_{\text{query}}^{\text{ref}}$, with EleutherAI/pythia-6.9b-deduped as the candidate model. Reference models of similar capability (like EleutherAI/pythia-6.9b or EleutherAI/pythia-2.8b) yield the lowest p-values.

Our approach follows a simple idea—given a prefix, we can sample multiple generations to estimate the next token probability. Moreover, Alice does not need to have an accurate probability estimation to compute the test statistics $\phi_{\text{query}}^{\text{ref}}$, as long as the ranking of the estimated probability is highly correlated with the ranking of the actual probability for each sequence. Specifically, we estimate the next token probability of prefix $x$ by random sampling model output $N$ times:

$$P(y_{gt}|x) \approx \mathbb{E}_{1 \le i \le N}\left[\mathbb{1}[y_{pred,i} = y_{gt}]\right]$$

where $y_{gt}$ is the ground truth continuation of $x$ in the transcript $\alpha$ and $y_{pred,i}$ is the predicted next token for the $i$th sample. Once we computed the estimated probability, We can apply a reference model as in Eq 2.

We conduct experiments on OLMo-7B, using 3M samples from the Epoch 1 transcript with a prefix length of 16. For each prefix, we repetitively sample up to 16 times. We use OLMo-7B-0724 as our reference model.

**Results.** We first verify that the Spearman correlation between the ranking of actual logprob and the ranking of estimated logprob is high. When the number of repeated samples is $N = 1/4/16$,

the correlation is $0.7068/0.8483/0.9165$ respectively. For our test using $\phi_{\text{query}}^{\text{ref}}$, the corresponding p-value is $1.6 \times 10^{-4}/5.6 \times 10^{-7}/6.4 \times 10^{-6}$.

# B  Observational Setting Experimental Details and Additional Results

## B.1  Implementation Details for $\phi_{\text{obs}}^{\text{part}}$

We build the index over a subset (18.75%) of n-grams of the `pythia-deduped` training set, up to the 100k training checkpoint. (Note that following our problem formulation, this n-gram index can be fully constructed from Alice's transcript (or subset of) $\alpha$.)

For each of Bob's texts in $x^{\beta}$, we count the number of n-gram matches with each of the training documents in our n-gram index, and keep a count of how many matches are at each training step. In particular, each $\mu_i$ is the list of n-grams seen at training step $i$ (e.g. between the $i$-th and $i+1$-th checkpoint) and the metric $\chi$ is the number of matches between $x^{\beta}$ and $\mu_i$. Since we index the first 100k checkpoints, we have the number of models $k = 100,000$. We count n-grams up to $n = 8$.

When counting n-grams, we first start with a series of 8 tokens, then decrease down to one token until a match is found. We also use a time-out feature for practicality, if searching the index for a particular n-gram takes more than 0.01 seconds on our machines, due to the high number of n-grams to index. We believe this does not impact the test effectiveness significantly.

## B.2  Implementation Details for $\phi_{\text{obs}}^{\text{shuff}}$

We treat Bob's text $x^{\beta}$ as a collection of shorter documents and let $\chi$ be the average log-likelihood of these documents under each model, either before or after finetuning. In the case where we do finetune on $x^{\beta}$, we do so for one epoch with a learning rate of $1 \times 10^{-5}$ with 4 documents per batch.

On TinyStories, we use the same model architecture ($\texttt{d\_model} = 256$, $\texttt{d\_ffn} = 512$, $\texttt{num\_layers} = 4$, approximately 3M parameters) as in the multiple epoch experiments. For the observational setting experiments, we train for a single epoch on 500K documents with a constant learning rate of $1 \times 10^{-5}$ and 4 documents per batch. We save checkpoints every 10k documents starting at 450K documents, which we use to resume training on reshuffled data to obtain the models $\mu_1, ..., \mu_k$ in our implementation of $\phi_{\text{obs}}^{\text{shuff}}$.

We obtain Bob's model by continuing to train on additional documents from the TinyStories training set (distinct from the 500K documents we use to train Alice's model). Unlike when we resume training from an intermediate checkpoint in $\phi_{\text{obs}}^{\text{shuff}}$, we reinitialize the optimizer state for Bob's model.

## B.3  Sampling Parameter Ablations for $\phi_{\text{obs}}^{\text{part}}$

We report p-values computed from $\phi_{\text{obs}}^{\text{part}}$ for the `pythia-6.9b-deduped-step100000` model and generated texts, which are prefixed with random sequences from The Pile. We vary sampling temperature and `top_p` (with temperature) in Figure 8.

We find that the results are somewhat sensitive to temperature; in particular, our test is less effective at lower temperatures, potentially due to the lack of text diversity limiting the effective sample size.

## B.4  TinyStories Ablations for $\phi_{\text{obs}}^{\text{shuff}}$

We report additional results on TinyStories with the $\phi_{\text{obs}}^{\text{shuff}}$ statistic. We vary the amount of retraining and finetuning together (Figure 9), the sampling temperature (Figure 10) and model size (Figure 11). We find that increasing the amount of retraining improves robustness to finetuning (as expected). Our test generally works well across a range of sampling temperatures, and notably works unexpectedly well at temperature 0.9 for enough tokens (recall we report results at temperature 1.0 for all other experiments). Finally, our test is more effective when scaling up the model size.

## B.5  Validity of $\phi_{\text{obs}}^{\text{shuff}}$

We plot the empirical distribution of output of $\phi_{\text{obs}}^{\text{shuff}}$ under the null in Figure 12. We generate samples of text from the null by training an independent copy of Alice's model from scratch (i.e., we keep all training hyperaparameters the same but use a different random seed). The output empirically behaves

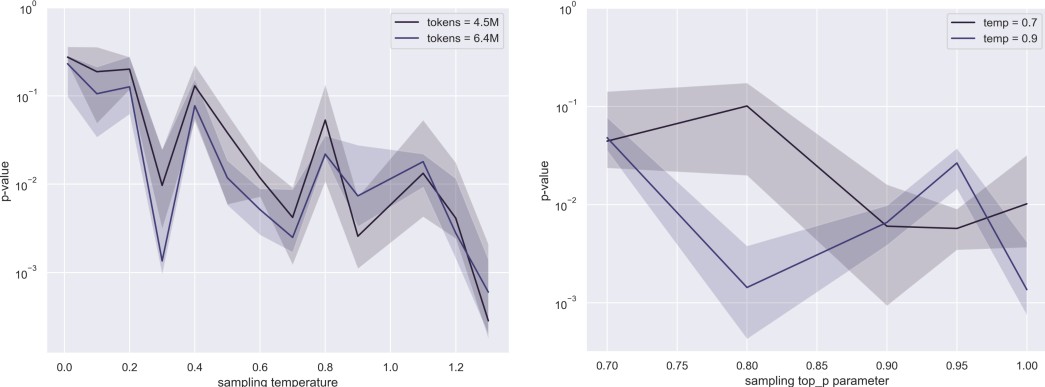

(a) We vary Bob's sampling temperature and number of total tokens.

(b) We vary Bob's top_p sampling parameter and fix the 6.4M total tokens.

Figure 8: We report median p-values and interquartile ranges over 10 trials of $\phi_{\text{obs}}^{\text{part}}$ using n-gram counts for `pythia-6.9b-deduped-step100000` while varying sampling parameters of Bob's model. Because sampling is expensive, for each trial we resample generations with replacement from an initial sample of 50K generations (i.e., 6.4M tokens), so the reported ranges are likely narrower than true confidence bands.

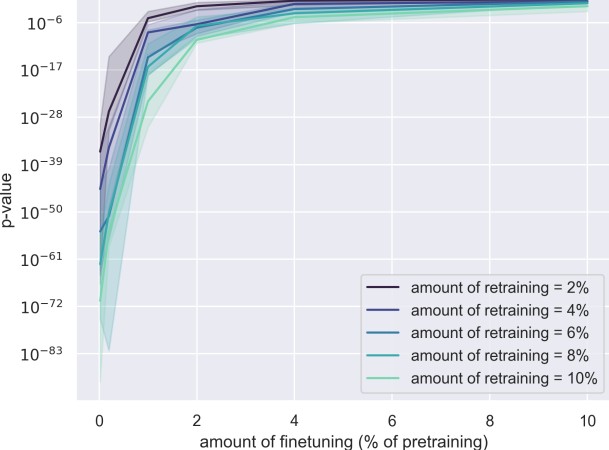

Figure 9: We report approximate p-values from applying $\phi_{\text{obs}}^{\text{shuff}}$ to TinyStories models, varying the amount of retraining in the test and the amount of finetuning by Bob.

like a conservative p-value, in the sense that the probability it is less than $x$ for $x \in [0, 1]$ is at most $x$ (whereas for a true p-value the probability would be equal to $x$). Concretely, this implies that the approximate p-values we report from $\phi_{\text{obs}}^{\text{shuff}}$ are typically larger than true p-values.

## B.6  Additional OLMo Results for $\phi_{\text{obs}}^{\text{shuff}}$

In Table 7, we evaluate $\phi_{\text{obs}}^{\text{shuff}}$ on `OLMo-2-0425-1B` and `OLMo-2-1124-7B` for different sizes of Bob's text, using the same protocol described in Section 4.3.2 but for smaller token counts. Notably, even evaluating a single generation (i.e., $|x^\beta| = 32$) we observe p-values on the order of $10^{-3}$ and $10^{-5}$ roughly a quarter of the time at the 1B and 7B scales respectively.

## C  Discussion of Related Work [17, 18]

We discuss two related works in the query setting. Both Nikolic et al. [17] and Jin et al. [18] try to determine whether two models are independently trained versus not based on the similarity of their outputs. Neither approach yields exact p-values, and moreover we show that they can be unreliable

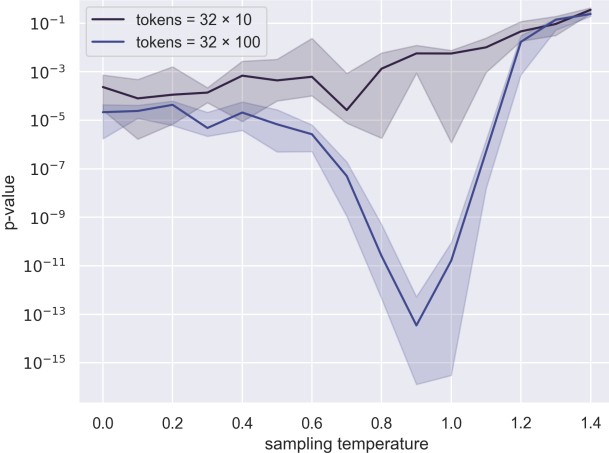

Figure 10: We report approximate p-values from applying $\phi_{\text{obs}}^{\text{shuff}}$ to TinyStories models, varying the sampling temperature of Bob's model.

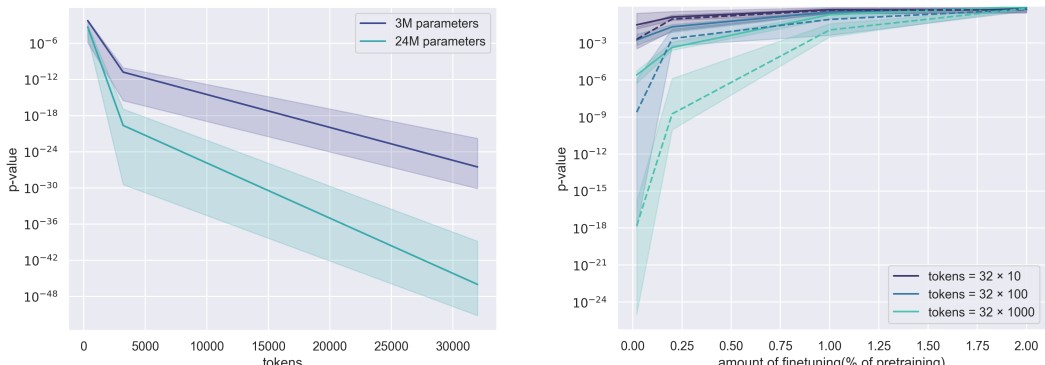

Figure 11: We report approximate p-values from applying $\phi_{\text{obs}}^{\text{shuff}}$ to TinyStories models of different scales. We obtain the 24M parameter model using the exact same procedure as for the 3M model, except we increase all intermediate activation sizes (i.e., both for self-attention and MLP layers) and the number of layers each by a factor of two. The total parameter counts are rough estimates. (Left) We vary the number of tokens in Bob's text and do not finetune on Bob's text. (Right) We finetune on Bob's text and vary the amount of finetuning by Bob, with results for the smaller model in solid.

in practice for models independently trained on similar data. By using specific prompts to generate responses, both methods also do not work for the observational setting.

Nikolic et al. [17] count token matches in outputs between two models $\mu^\alpha$ and $\mu^\beta$ (i.e., Alice and Bob's models) over prompts $x^j$, and compare a similarity score based on these counts between $\mu^\alpha$ and $\mu^\beta$ versus between $\mu^\beta$ and a class of reference models (see Algorithm 1 in [17]). In particular, this does not yield provably exact p-values for independently trained models.

We run an experiment involving Pythia models to show a failure mode of their test—the test indicates that `pythia-1.4b` and `pythia-1.4b-deduped` are *not* independent, even though they were two independent runs trained on two different transcripts. Specifically, we use the reference models given in their Bench A and compute similarity of Pile sequence prompts (which likely all the models were trained on) with `pythia-1.4b`, shown in Table 8.

We see that the similarity between the independent `pythia-1.4b` and `pythia-1.4b-deduped` models (0.7976) is higher than the similarity between the finetune pair `pythia-1.4b` and `pythia1.4B-finetuned-on-lamini-docs` (0.7714). Using a z-test with the given reference models from [17] would yield a p-value of 9.622e-06 for the independence between `pythia-1.4b` and `pythia-1.4b-deduped`, when in fact they are independent, meaning there is potentially a very

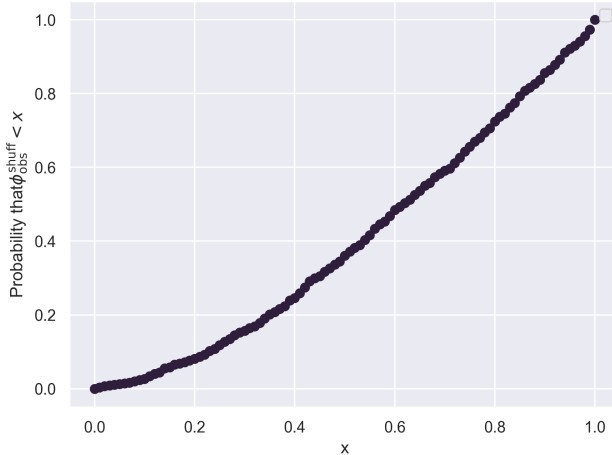

Figure 12: We plot the distribution of the output by $\phi_{\text{obs}}^{\text{shuff}}$ under the null.

| Model Size | $|x^\beta| = 32$ | 96 | 160 | 320 |
|---|---|---|---|---|
| 1B | $1.7 \times 10^{-1}$ | $9.7 \times 10^{-2}$ | $1.1 \times 10^{-1}$ | $1.3 \times 10^{-1}$ |
| | $(4.4 \times 10^{-3}, 4.6 \times 10^{-1})$ | $(1.0 \times 10^{-6}, 3.3 \times 10^{-1})$ | $(3.4 \times 10^{-5}, 2.2 \times 10^{-1})$ | $(6.5 \times 10^{-3}, 2.8 \times 10^{-1})$ |
| 7B | $3.7 \times 10^{-1}$ | $1.3 \times 10^{-1}$ | $9.9 \times 10^{-2}$ | $4.5 \times 10^{-3}$ |
| | $(2.7 \times 10^{-5}, 7.5 \times 10^{-1})$ | $(5.4 \times 10^{-3}, 3.0 \times 10^{-1})$ | $(3.0 \times 10^{-5}, 3.2 \times 10^{-1})$ | $(1.0 \times 10^{-6}, 4.9 \times 10^{-2})$ |

Table 7: Median p-values (with interquartile ranges) across 10 trials from applying $\phi_{\text{obs}}^{\text{shuff}}$ to `OLMo-2-0425-1B` and `OLMo-2-1124-7B`.

high false positive rate. In contrast, our methods yield provably exact p-values (see Theorem 1) where we can control for Type-I error.

Jin et al. [18] craft prompts for models and compare the similarity of outputs. Like Nikolic et al. [17], they use a metric "target response rate" (TRR) that measures similarity of responses, and they identify two models as fine-tunes if their TRR is higher compared to the values for non fine-tunes. However, they find that independently trained models from the same model developer may have similar responses, and they classify such models as "related" (e.g. may share training data, for example). As such, in Table III of their paper, they find the TRR between `Mistral-7B-v0.1` and `Mistral-7B-v0.2` is 0.70 and higher than the TRR with any `Mistral-7B-v0.1` fine-tunes, although the first two are independent.

# D    Compute and Cost

## D.1    Audits and Cost Analysis

**Auditing and cost.** We give cost estimates of running the query-setting test for current language model APIs. For our test under the query setting, we observe that logprob over 64M token samples is typically enough to achieve a p-value below $10^{-3}$. Thus, we estimate the cost using a sample of 8M sequences, each with 8 tokens.

As of May 2025, the cost of querying 1M inference tokens for OpenAI GPT-4.1 mini is $0.40 per 1M input tokens and $1.60 per 1M output tokens[7]. Computing an average over a window size of 7 ($8 \times 8/2 = 32$ tokens per sequence) with one output token per window would have a cost of $0.40 \times 8 \times (1 + 2 + \cdots + 7) + \$1.60 \times 8 \times 7 =$179.20. Some models may require fewer tokens, such as the Pythia family.

Note that some APIs only give top-k logprobs or text output. In this case, one can estimate logprobs using the method described in Appendix A.6.

---

[7]https://platform.openai.com/docs/pricing

| model | fine-tuned? | $\mu$ with `pythia-1.4b` |
|---|:---:|:---:|
| `meta-llama/Llama-3.2-1B-Instruct` | × | 0.5451 |
| `meta-llama/Llama-3.2-3B-Instruct` | × | 0.5408 |
| `microsoft/Phi-3-mini-4k-instruct` | × | 0.6137 |
| `microsoft/phi-2` | × | 0.6453 |
| `google/gemma-2b` | × | 0.6322 |
| `google/gemma-2-2b` | × | 0.6148 |
| `Qwen/Qwen2-1.5B` | × | 0.6126 |
| `Qwen/Qwen2.5-1.5B-Instruct` | × | 0.4875 |
| `deepseek-ai/deepseek-coder-1.3b-base` | × | 0.5473 |
| `TinyLlama/TinyLlama-1.1B-Chat-v1.0` | × | 0.6050 |
| `EleutherAI/pythia-1.4b` | ✓ | 1.0000 |
| `nnheui/pythia-1.4b-sft-full` | ✓ | 0.8205 |
| `herMaster/pythia1.4B-finetuned-on-lamini-docs` | ✓ | 0.7714 |
| `EleutherAI/pythia-1.4b-deduped` | × | 0.7976 |

Table 8: We compute model output similarities $\mu$ following Nikolic et al. [17] between `pythia-1.4b` and other models on HuggingFace. The similarity with the independent `pythia-1.4b-deduped` is higher than the similarity with a `pythia-1.4b` fine-tune.

The observational setting does not require querying an API, as Alice passively observes text generated by Bob's model.

### D.2 Compute Resources

We run our experiments on an internal cluster using NVIDIA A100 and A6000 GPUs. The primary compute use for our methods is getting the perplexity of samples for different models. For a 6.9b-parameter language model (the majority of our experiments), computing the perplexity on 1M sequences of length 64 tokens per sequence takes around 75 mins on a single A100 GPU.

For the observational setting, we build the n-gram to training steps index for Pythia's pre-training data using the infini-gram package [42]. We index 18.75% of the training data, which takes 258G disk space. To run the test $\phi_{\text{obs}}^{\text{part}}$ once all the documents are processed, which involves getting the counts of n-gram matches at each train step, takes around 1 minute for 100000 texts (12.8M tokens), and scales approximately linearly with number of tokens (most processing time is disk IO).

Training the TinyStories models described in Appendix A.4 for 50000 sequences [40] takes around 10 minutes on a A6000 machine.

