# OpenReview forum: "Blackbox Model Provenance via Palimpsestic Membership Inference"
_NeurIPS.cc/2025/Conference — NeurIPS 2025 spotlight_

### Official Review · Reviewer_UXNh · 2025-06-28

**Clarity:** 2
**Significance:** 3
**Originality:** 3
**Rating:** 4
**Confidence:** 4

**Summary:**

This paper proposes tests for verifying the provenance of open-weight language models, based on the hypothesis that data seen later in training is more likely to be memorized. It includes tests for both query and sampling settings and demonstrates the effectiveness of these tests on several language models.

**Questions:**

1. When using pre-trained models as Alice's models, how to obtain the training transcript (i.e., the order of training samples)?

2. In the sampling settings, how does the temperature parameter in Bob's model affect the results?

**Ethical Concerns:**

["NO or VERY MINOR ethics concerns only"]

**Final Justification:**

The rebuttal addressed some of my concerns. However, my concern about its practicality remains, particularly due to the requirement of access to training data. Additionally, the clarity of the paper should be improved. Thus, I’ll raise my rating to a 4 but not higher.

**Limitations:**

Yes

**Quality:**

2

**Strengths And Weaknesses:**

**Strengths:**

1. The proposed method does not require modifying the model or inserting special triggers.

2. The tests are evaluated on several families of language models.

**Weaknesses:**

1. The clarity of the paper needs improvement. For example, Definition 2 is confusing.

2. Lack of membership inference context: Although “membership inference” is used in the title, the paper provides no discussion of its relationship to prior membership inference attacks. If the term is meant differently, this should be clarified; otherwise, either background should be added or the term removed from the title.

3. Algorithm 2 is problematic (single_kgrams is not defined).

4. The method assumes that the model developer has access to the exact training order and can expose (at least a part of) the training data. This may be impractical in real-world settings, especially for large-scale training runs where streaming, multi-phase data loading, or privacy constraints prevent full transcript reconstruction.

5.  The paper does not explore how Bob’s decoding strategy (e.g., sampling temperature) affects the test performance.
---

---

> ### Author Rebuttal · Authors · 2025-07-31
>
> We thank the reviewer for their time and effort. We address their questions and concerns below.
>
> **Weakness 1** (Definition 2)
>
> Definition 2 aims to establish our primary assumption that the training algorithm $A$ shuffles the data order, which we use for the following algorithms. We will clarify this definition in the revision.
>
> **Weakness 2** (Membership inference context)
>
> Thanks for the suggestion! We acknowledge that in the literature, “membership inference” refers to predicting whether an individual example is in the training dataset. Here, we are extending the definition following [1, 2]: instead of focusing on the membership of an individual example, or the membership of a set of examples (e.g., dataset inference), we consider the “palimpsestic membership” as whether "a sequence of examples" is a subsequence of the training examples. We agree that this is not the standard implication of membership inference. We will add more explanation for this in the revision.
>
> Our work is also related to membership inference attacks. The features we use are commonly used in membership inference attacks, e.g., logprobs and logprobs with a reference. We believe better features discovered by the membership inference community could also help improve our test.
>
>
> **Weakness 3** (Typo)
>
> We apologize for the typo; `single_kgrams` should be `matches`.
>
> **Weakness 4** (Exact training order in real-world settings)
>
> We agree that reconstructing and releasing the full transcript might not be feasible in practice. However, we would like to clarify that our method only requires knowing the relative order of a small sample (<0.0001%) of the training data, and Alice does not need to release the full copy of her training data publicly. Specifically, for technical constraints such as data streaming and multi-phase loading, as long as Alice can track the relative order of a small subset of data, for example, logging a few thousands training examples once in a while, she can run this test. For privacy constraints, disclosing a full copy of training data is only required for public verification under the query setting, in which case Alice can choose to release a subset from a commonly used, CC licensed corpus, such as Wikitext (also see our response to Reviewer zdZQ, weakness 1).
>
> **Weakness 5** (Sampling strategies for the sample setting)
>
> Please refer to our response to question 2.
>
> **Question 1** (Obtaining the training transcript for pretrained models)
>
> For Pythia and OLMo, the training transcript is reconstructed using a global index, i.e., a mapping from training example order to the order in the original memory-mapped dataset. This global index is either directly provided by the model developers, or can be deterministically regenerated from the random seed and the total number of examples through a shuffle, ensuring the exact training order is reproducible.
>
> **Question 2** (Varying Bob's sampling strategies (temperature and top_p) for the sample setting)
>
> We present additional experiments varying decoding strategies (temperature and nucleus sampling) and the prompt distribution in the sample setting and present the p-values. These are using a different method from the paper; here, we count the number of n-gram matches from each training step, and take the Spearman correlation between the number of matches and the training step. (In Algorithm 2, this would be taking a frequency count (like `collections.Counter`) over `matches`, and taking the Spearman rank with `arange(100000)`). Using the Spearman correlation this way is valid for the same reason it is valid for the query setting.
>
> We vary the temperature for Pile completions for the `pythia-6.9b-deduped` model, compared with itself, and keep `top_p`=1.0. With lower temperatures, we find the p-values are larger, but relatively low. At 50000 texts (each 128 tokens), the results are relatively stable across temperatures. We then vary `top_p` and find similar results. Sometimes the p-values can be noisy, so in the revision we will report distributions of p-values for different samples of $N$ texts.
>
> p-values, varying temperature:
> | temp | N=5000 | 10000 | 20000 | 50000 |
> | --- | -- | -- | -- | -- |
> | 0.01 | 0.711 | 0.685 | 0.033 | 0.072 |
> | 0.1 | 0.831 | 0.888 | 0.021 | 0.062 |
> | 0.2 | 0.886 | 0.159 | 0.534 | 0.072 |
> | 0.3 | 0.015 | 0.103 | 0.004 | 0.0007 |
> | 0.4 | 0.858 | 0.144 | 0.149 | 0.032 |
> | 0.5 | 0.038 | 0.002 | 0.054 | 0.002 |
> | 0.6 | 0.023 | 0.422 | 0.007 | 0.0005 |
> | 0.7 | 0.002 | 0.537 | 0.013 | 0.0002 |
> | 0.8 | 0.124 | 0.315 | 0.010 | 0.011 |
> | 0.9 | 0.007 | 0.003 | 0.074 | 0.001 |
> | 1.1 | 0.113 | 0.028 | 0.0007 | 0.0009 |
> | 1.2 | 0.030 | 0.218 | 0.041 | 0.0001 |
> | 1.3 | 0.008 | 0.003 | 0.004 | 7.55e-06 |
>
> p-values, varying `top_p`:
> | temp | top_p | N=5000 | 10000 | 20000 | 50000 |
> | --- | --- | --- | --- | --- | --- |
> | 0.7 | 1.0 | 0.379 | 0.532 | 0.235 | 0.001 |
> | 0.7 | 0.95 | 0.647 | 0.309 | 0.154 | 0.001 |
> | 0.7 | 0.9 | 0.701 | 0.194 | 0.120 | 0.0006 |
> | 0.7 | 0.8 | 0.281 | 0.032 | 0.010 | 0.018 |
> | 0.7 | 0.7 | 0.073 | 0.054 | 0.039 | 0.011 |
> | 0.9 | 1.0 | 0.509 | 0.152 | 0.046 | 0.0005 |
> | 0.9 | 0.95 | 0.416 | 0.460 | 0.005 | 0.005 |
> | 0.9 | 0.9 | 0.021 | 0.028 | 0.003 | 0.001 |
> | 0.9 | 0.8 | 0.754 | 0.364 | 0.067 | 2.29e-05 |
> | 0.9 | 0.7 | 0.613 | 0.734 | 0.563 | 0.029 |
>
> We also try varying the prompt distribution by sampling prompts from Wikitext (also for the `pythia-6.9b-deduped` model). We report the p-values below and find they are also significant at 50000 texts.
> | temp | N=5000 | 10000 | 25000 | 50000 |
> | --- | --- | --- | --- | --- |
> | 0.8 | 0.401 | 0.158 | 0.0267 | 0.0032 | 0.0011 |
>
> (For a sanity check, these are the p-values for using completions from the `pythia-6.9b` model compared with the `pythia-6.9b-deduped` training order, which we see are not significant.)
> | temp | N=5000 | 10000 | 25000 | 50000 |
> | --- | --- | --- | --- | --- |
> | 0.8 | 0.600 | 0.310 | 0.659 | 0.544 | 0.664 |
>
> We appreciate the reviewer's comments and are happy to discuss further!
>
> [1] Pratyush Maini, Mohammad Yaghini, and Nicolas Papernot. Dataset inference: Ownership resolution in machine learning. In International Conference on Learning Representations, 2021.
>
> [2] Pratyush Maini, Hengrui Jia, Nicolas Papernot, Adam Dziedzic. LLM Dataset Inference Did you train on my dataset? In The Thirty-eighth Annual Conference on Neural Information Processing Systems, 2024.

---

> > ### Comment · Reviewer_UXNh · 2025-08-04
> >
> > Thank you for the rebuttal. It addressed some of my concerns; however, my reservations about the practicality of the approach still remain. Overall, I’ll raise my rating to a 4.

---

### Official Review · Reviewer_AR86 · 2025-07-01

**Clarity:** 3
**Significance:** 3
**Originality:** 4
**Rating:** 5
**Confidence:** 4

**Summary:**

The goal of the paper is to give a statistical test that allows one to prove that some model B was derived from their model A. (Derivation here may be unintentional, and can take many forms, such as training on the model’s outputs or fine-tuning the original model.) Although this task is not new, the paper proposes a novel method to obtain control over the null distribution: the observation that training runs use random permutations to decide their data ordering. This allows one to perform this test without making any modifications to the data or training of the model. Indeed, the test can be applied to models that were trained in the past, so long as a record the data ordering used is available.

**Questions:**

1. It is common for the best models to be trained for multiple epochs on the same data. Do you think the method will continue to work in that setting?

**Ethical Concerns:**

["NO or VERY MINOR ethics concerns only"]

**Final Justification:**

I think the paper presents a solid approach to an important task (dataset inference) which has which leverages a nearly universal source of null hypothesis (randomized of data order) and obtains good results. The authors responded well to my questions as well.

**Limitations:**

See weakness 1.

**Quality:**

3

**Strengths And Weaknesses:**

## Strengths

1. The approach leveraging the data ordering is very neat and perfectly suitable for the task at hand. It makes provenance testing usable in many more situations than previous methods (e.g. dataset inference requires an IID hold out set which are not available for many popular datasets.)
2. The paper reports results for both Pythia and OLMo models, the two most prominent models where the training data order is known.

## Weaknesses

1.  The main weakness is that the very idea of “model provenance” is a bit overly broad and it’s not clear how useful it will be in practice. A similar idea is articulated in [1], but essentially the idea is that any model B which is trained on data written or generated after the release of model A is very unlikely to be truly independent of it. For example humans may read the output of A, influencing their style of writing, and thus a model B trained on text written by them would not be independent of A. To be clear, I think the paper is interesting and impactful regardless but I do think some discussion of this issue would be nice to have.

[1] Position: Membership Inference Attacks Cannot Prove that a Model Was Trained on Your Data.” Jie Zhang, Debeshee Das, Gautam Kamath, Florian Tramèr. IEEE SaTML 2025 / arXiv:2409.19798

---

> ### Author Rebuttal · Authors · 2025-07-31
>
> We thank the reviewer for their time and effort. We address their questions and concerns below.
>
> **Weakness 1** (Discussion of model provenance)
>
> This is an excellent point — over time, it is possible that model A could influence model B (or human text that it is trained on) indirectly, and we agree that this is relevant discussion for "model provenance". The main issue [1] raises is a lack of a clear null hypothesis with respect to training data composition. Crucially, our methods are valid regardless of dataset composition, and we have a clear null hypothesis (independence with A’s training order).
>
> We are careful not to claim that our methods can show Bob has necessarily fine-tuned Alice’s model. It is true that if our method labels a model and set of texts as independent, it’s still possible they share some other provenance (not “truly independent”). In the future, one can imagine model providers may run our tests on large samples of human text in order to re-establish a (approximate) null baseline. We will include this discussion in the revision.
>
> **Question 1** (Multiple epochs of training)
>
> Unfortunately we could not test this for large models trained for multiple epochs as none are fully open-source (OLMo and Pythia models are trained for up to 1.5 epochs). We show that at least after an additional 0.5 epochs, we can still use the first-epoch order to get significant p-values (Table 2 in the appendix), and there are strong correlations with the last four epochs’ orders for TinyStories models (Appendix A.4). In practice, one could use the training order of the final epoch for the best signal and we believe our method would still work in that setting.
>
> We appreciate the reviewer's comments and are happy to discuss further!

---

> > ### Comment · Reviewer_AR86 · 2025-08-03
> >
> > I would like to thank the authors for answering my questions. I appreciate their well-considered approach to the subtle topic of provenance. I am keeping my (high) score and I would like to reaffirm that I think this work should be accepted.

---

### Official Review · Reviewer_Gt7T · 2025-07-02

**Clarity:** 4
**Significance:** 4
**Originality:** 4
**Rating:** 5
**Confidence:** 3

**Summary:**

This paper studies the following question: Suppose one party, Alice, trains an open-weight language model, and a second party, Bob, produces text using a derivative of that model. Can Alive prove that Bob is using her model? This question may be important for a company like Meta, whose terms of service requires that those using their models within a product would need to acknowledge that the product is "built with Meta Llama 3."

The main insight of the authors' approach is the observation that LMs are typically trained on randomly shuffled data, and **data seen later in training is more likely to be memorized**. Therefore, we can test whether Bob's model derives from Alice by correlating its output with the ordering of examples in Alice's training run. The test works in two settings:
1. **The query setting:** Given the ability to obtain log-likelihoods from Bob's model for a given piece of text, test whether the log-likelihoods assigned to pretraining examples correlates with the order of those examples in pretraining.
2. **The sample setting:** Given only the ability to sample text from Bob's model, test whether n-gram overlap between the generations and pretraining examples correlates with the order of those examples in pretraining.

**Questions:**

1. Why is transparency important for a good test? The authors argue this is a downside of tests based on holding out a secret test set or embedding a secret fingerprint, but it is not clear why model developers would prefer completely transparent tests.
2. Can you comment more on the training data for each of the models you studied? E.g., Is it known to be true that `pythia-deduped` reshuffles the (deduplicated) training data of `pythia`, or is it possible that `pythia-deduped` "pulls out" the duplicated examples from the current ordering?
3. In the caption of Figure 1c, what does "*The order of losses matches the training order*" mean? Do you mean loss diffs at the end of training for both models? In general, the interpretation of the results in this caption versus §4.2 can be cleaned up. It is not always clear how the plots should be interpreted to support all of the claims.
4. In the sample setting, does the test work at detecting post-trains of Alice's model? If I understand correctly, it's using the same `pythia-deduped` model as both Alice's and Bob's model (unlike the query setting).
5. Are `OLMo-0424` and `OLMo-0724` both used?

**Other suggestions:**
- It may be a good idea to define more of the notation used in §3.1, e.g., $\Delta$, $\perp$ (since I assume here it does not indicate difference or perpendicular to)
- L. 259: "*earses*" $\rightarrow$ "*erases*"
- Since the TinyStories results are only shown in the Appendix, I would recommend removing their mentions in the main paper. At the end of the paper, I was confused about not seeing any results on TinyStories despite it being mentioned multiple times earlier in the paper.

**Ethical Concerns:**

["NO or VERY MINOR ethics concerns only"]

**Limitations:**

Yes.

**Quality:**

4

**Strengths And Weaknesses:**

### Strengths:
1. The paper describes the problem setup very well, and clearly distinguishes multiple lines of related work and their different assumptions and realms of applicability. Beyond addressing this particular problem, the paper is rife with insights that are interesting to the broader memorization community.
2. The paper presents an extremely effective and clever solution, based on the insight that models assign higher likelihood to text seen later in pretraining. In the query setting, it achieves essentially perfect precision and recall at detecting model derivatives.
3. The test works without requiring Alice to hold out *any* information about the test, unlike methods which use a secret held-out test set or embed secret strings into the pretraining data.

### Weaknesses:
1. Experiments in the sample setting are much more limited than those in the query setting. For instance, if I understand correctly, the sample setting doesn't actually consider detectin model *derivatives*. While it is expected for the test to be less effective in the sample setting (due to Alice having less information to work with), it would provide valuable insights if the experimental settings were more comparable.
2. Ultimately, if Bob is hosting (a derivative of) Alice's model through an API, the query setting (where users can access log probabilities) is pretty unrealistic.

---

> ### Author Rebuttal · Authors · 2025-07-31
>
> We thank the reviewer for their time and effort. We address their questions and concerns below.
>
> **Question 1** (Transparency)
>
> This is a good question. Transparency is important so that the test can be used by third-parties and auditors, for example. It is also important if a model developer (Alice) wants to prove that Bob’s model is a derivative —- she must disclose the secret test set or fingerprint so another party can independently verify the result.
>
> **Question 2** (Pythia training data)
>
> In general, Pythia and OLMo construct the training data by first filtering (include deduplication) and then apply a global shuffle. As a result, the training data of `pythia-deduped` is shuffled independently of `pythia`. The implementation details can be found in the pythia and OLMo git repositories.
>
> We use the training scripts provided by EleutherAI, in which they provide the seeds. Pythia and OLMo generate the training data by filtering / deduplicating then shuffling, so the `pythia-deduped` and `pythia` datasets are shuffled independently.
>
> **Question 3** (Figure 1c clarification)
>
> Yes, we mean the differences in losses (e.g. subtracting the loss of a reference model), although the order can also be seen in 1(a), which is just the loss. These plots are mostly to demonstrate that the model in fact preserves the training order, which is more visible when using a reference model. We will add more explanation in the revision.
>
> **Question 4** (Fine-tuned models for the sample setting)
>
> Yes, our current experiments in the paper are for the same model for Alice and Bob; we provide additional experiments for the fine-tuned setting in the discussion for Weakness 1.
>
> **Question 5** (Are OLMo-0424 and OLMo-0724 both used?)
>
> Yes, in Figure 2 (b) in the main text, Table 3, and Table 4 in the appendix, we show results for fine-tunes of both `OLMo-0424` and `OLMo-0724`.
>
> **Weakness 1** (Additional experiments in the sample setting for fine-tuned models)
>
> We provide some additional experiments for fine-tuned models, specifically for later checkpoints of the `pythia-6.9b-deduped` model. These are using slightly a different method from the paper, which we found it’s more effective; here, we count the number of n-gram matches from each training step, and take the Spearman correlation between the number of matches and the training step. (In Algorithm 2, this would be taking a frequency count (like `collections.Counter`) over “matches”, and taking the Spearman rank with `arange(100000)`). Using the Spearman correlation this way is valid for the same reason it is valid for the query setting. We use the base model `pythia-6.9b-deduped` at checkpoint 100000 (Alice) and generate $N$ texts (Bob’s texts) from checkpoints 105000, 110000, 115000, 120000, and 130000.
>
> We find that we get low p-values even after fine-tuning for 30% the number of pretraining tokens, if we use enough generated texts.
>
> | checkpoint | N=5000 | 10000 | 25000 | 50000 | 100000 | 200000 |
> | -------------| --------|---------|-----------|-----------|------------|-----------|
> | step100000 | 0.0196 | 0.0128 | 0.00136 | 0.00063 | 4.95e-05 | 1.51e-05 |
> | step105000 | 0.128 | 0.0106 | 0.0400 | 0.0196 | 0.0013 | 0.00451 |
> | step110000 | 0.312 | 0.206 | 0.238 | 0.0906 | 0.0575 | 0.0190 |
> | step115000 | 0.575 | 0.0455 | 0.0117 | 0.0179 | 0.0506 | 0.0863 |
> | step120000 | 0.819 | 0.940 | 0.478 | 0.450 | 0.185 | 0.0305 |
> | step130000 | 0.325 | 0.380 | 0.214 | 0.0638 | 0.0852 | 0.0406 |
>
> **Weakness 2** (API logprobs)
>
> This is true, but logprobs can be estimated from an API with repeated sampling (we show this is feasible under a reasonable budget in Appendix A.6 and C.2), or one could use the sample setting.
>
> We also thank the reviewer for their additional suggestions and will implement them in the revision. We appreciate the reviewer's engagement and are happy to discuss further!

---

> > ### Comment · Reviewer_Gt7T · 2025-08-09
> >
> > Thank you for the interesting response. I will maintain my positive score.

---

### Official Review · Reviewer_zdZQ · 2025-07-06

**Clarity:** 3
**Significance:** 3
**Originality:** 3
**Rating:** 4
**Confidence:** 3

**Summary:**

This paper introduces a novel black-box model provenance method by analyzing the memory properties of a language model on the order of its training data. The method proposes tests in two settings: query setting and sample setting, and empirically verifies its effectiveness even when the model has undergone common post-training practices. This work provides a new framework for provenance verification of open source language models, which can help improve model accountability and protection.

**Questions:**

In experiments 4.3.1, why samples the training data from the first epoch to construct the transcript?

**Ethical Concerns:**

["NO or VERY MINOR ethics concerns only"]

**Final Justification:**

The reviewer's rebuttal addressed most of my concerns, but I still have concerns about the practicality.

**Limitations:**

The threat model should be further discussed. What information Alice has to disclose? How about prior work? What information should be disclosed in their settings? This is important because the proposed method looks like a new framework with new threat model.

**Quality:**

2

**Strengths And Weaknesses:**

Strengths:
1. This paper proposes a unique method based on "palindromic memory" to solve the model traceability problem by using the language model's memory property of the training data sequence. This is different from previous methods that rely on private triggers or test sets and is innovative.
2. Wide applicability and robustness: The method is robust to common post-training practices (such as supervised fine-tuning, preference optimization, model fusion, etc.) in the query setting, and can reliably attribute text with only thousands of tokens in the sample setting, which greatly broadens its potential for real-world applications.
3. Proposed testing framework is able to provide provable false positive error control (via precise p-values), a significant advantage over existing work that relies on heuristics, providing statistically rigorous guarantees on model provenance.

Weakness:
1. The main limitation of this method is that it requires access to Alice's training data to perform testing. Although the authors mention that it is possible to use only a subset of the training data, in practice model developers may be reluctant to disclose their training data. While prior work like [1] do not need to public the training data, they only public the hash keys for verification. In the experiments, Alice expose 1M training samples to construct the transcript. Which is a large privacy concern, especially considering prior work does not need to leak any training data.
2. If Bob has the transcript, it can shuffle the training dataset and fine-tune the model on such adversaial desined trainscript. So the memory of the LLMs will be shuffled. In this case, can the proposed test can identify such fine-tuned LLMs? If so, why? Since the insight of this paper is that the later memories will overwrite earlier memories, so if the attacker also uses this to fine-tune, wouldn't it be possible to eliminate the previous order?
3. For smaller models (e.g., Pythia-1.4b), the test showed false negatives in some cases, which may be due to "catastrophic forgetting" causing the model's performance on the pre-training data to drop significantly after fine-tuning. This means that the method may not be effective in identifying fine-tuned models derived from the Alice model in some cases, especially when the model quality is severely affected.

[1] A Watermark for Large Language Models.

---

> ### Author Rebuttal · Authors · 2025-07-31
>
> We thank the reviewer for their time and effort. We address their questions and concerns below.
>
> **Weakness 1** (Disclosure of Alice's training data)
>
> We understand the reviewer's concerns on disclosing information about the training data. We would like to further clarify when and what Alice needs to disclose about her training data.
>
> Publishing information of training data is only needed **for third parties to verify the test result**. The main use case of this test is for the model developer (Alice), who clearly has her full training data, to check whether someone else has used their model. In this case, Alice can keep her training data private.
>
> If she wishes to prove to someone else that others are using her model (for public verifiability purposes), she can publish the order of a dataset without privacy concerns like Wikitext. Specifically, for public verification,
>
> - under the query setting, Alice needs to disclose usually less than 1M sequences of her training data. As we discuss on L323-324, Alice can choose which subset of the training data to make public. For example, she may choose to release Wikitext samples. Wikitext is used in training nearly all language models and is under the Creative Commons license, which would not be a privacy concern.
>
> - **under the sample setting, Alice does not need to disclose any exact copy of her training data**. She only needs to release aggregated n-gram stats of the data for some small n. Given the nature of information that needs to be disclosed, we believe the privacy risk can be mitigated (especially if the subset is something like Common Crawl where releasing it would not have privacy concerns).
>
> Prior methods like [1] have not considered this public verifiability perspective, which is why they do not need to publish test implementation details. Also, we would like to emphasize that inference-time watermarking methods like [1] are not applicable to the sample setting, as Bob can directly sample text himself without the watermark (L112-L114).
>
> **Weakness 2** (Retraining on the shuffled data)
>
> This is indeed an interesting question! We have studied this adversarial setting in Section 4.3.3. In our experiments in TinyStories models (Appendix A.4), we find retraining on the same texts with a different shuffle still preserves the orders of the recent iterations. Specifically, we tried retraining for 10 epochs on TinyStories, and the orderings from the last 4 epochs **all** yielded significant correlation with the final model (Figure 6 of the appendix). We agree that with enough epoching over the same texts, the original order could be lost, however, this means to fully mask Alice’s original training order, Bob would have to do fine-tuning at the scale of the original pretraining budget. In fact, on Pythia and OLMo models, we find that even after retraining 50% of the original pretraining texts (L289-L291 and Figure 5), our tests still yield extremely low p-values. Moreover, fine-tuning on a small subset of the transcript, such as one million sequences, is not sufficient to evade the test. In that case, Alice can use another random subset of her training data, which very likely Bob has not retrained on.
>
> Regarding memorization, we would like to clarify that the most interesting finding is that, in the multi-epoch setting, earlier pretraining orders are inscribed into the model along with later ones (L292-L293), instead of being completely overwritten. This is exactly why we call it **“palimpsestic” memorization**.
>
> **Weakness 3** (Catastrophic forgetting in smaller models)
>
> As the reviewer noted, we show that all failure cases have much higher loss on general language modeling, which means the fine-tuned model’s quality is severely affected. Doing such fine-tuning defeats the purpose of modeling stealing, as these fine-tuned models would not be very usable in practice.
>
> Moreover, larger models are the most likely target of model stealing and misuse, and our test yields low p-values on all 7B and 12B models tested. This is in line with existing studies that show smaller models memorize less [2, 3].
>
> [2] Nicholas Carlini, Daphne Ippolito, Matthew Jagielski, Katherine Lee, Florian Tramer, and Chiyuan Zhang. 2023. Quantifying memorization across neural language models. In The Eleventh International Conference on Learning Representations.
>
> [3] Pietro Lesci, Clara Meister, Thomas Hofmann, Andreas Vlachos, and Tiago Pimentel. 2024. Causal estimation of memorisation profiles.
>
> **Question** (Sampling training data from the first epoch in experiments)
>
> We sample from the first epoch for consistency across model families (for example, `pythia` models are trained for one epoch, whereas `pythia-deduped` models are trained for more than one epoch). The fact that our tests yield such results even using only the earliest epoch ordering shows how effective it is against continued pre-training or fine-tuning.
>
> **Limitation** (Prior work and disclosure of Alice's training data)
>
> To run this test, Alice would have to disclose a subset of her training data (discussed in weakness 1). However, this subset could be something like Wikitext that would not be proprietary. The most-relevant prior work is [4] which requires using a held-out validation set that would also have to be disclosed to run the test. In particular, this validation set has to be i.i.d., i.e., must be representative of the entire training data, and would in fact be more proprietary.
>
> [4] Pratyush Maini, Mohammad Yaghini, and Nicolas Papernot. Dataset inference: Ownership resolution in machine learning. In International Conference on Learning Representations, 2021.
>
> We appreciate the reviewer's comments and are happy to discuss further!

---

> ### Comment · Reviewer_zdZQ · 2025-08-04
>
> Thank you for the authors’ detailed responses. After carefully reading through the paper and other reviewers’ comments, I still have several concerns that I hope the authors can address further:
>
> 1. **Practicality of Sampling Requirements**
>    The proposed method requires a large amount of text (often exceeding $10^5$ tokens) for effective detection. In realistic scenarios, obtaining such a volume of *pure* samples originating solely from a single LLM is often impractical.
>
>    * For example, in many real-world cases, an adversary (Bob) may generate content using multiple LLMs, mixing outputs from Alice’s LLM with other LLMs. In such mixed-generation settings, how does the method ensure that the statistical correlation attributed to Alice’s training order is not diluted or masked by contributions from other LLMs?
>    * Additionally, in the sample setting, Alice cannot guarantee that the collected text spans are all generated exclusively by her LLM. A common scenario could involve Bob using Alice's LLM as part of a novel-writing pipeline that includes outputs from various LLMs. And the Bob can adopt paraphasing attack on the generated texts. How would the proposed method handle common scenario?
>    * Similarly, in query-based settings, Bob could employ a mixture-of-experts (MoE) setup, dispatching different queries to different LLMs (including Alice’s). How robust is the proposed independence test in these adversarial hybrid-LLM configurations?
>    * Furthermore, how would the approach handle situations where Bob blends Alice’s LLM with independently trained LLMs (e.g., via LLM merging or weight averaging)? Does the detection power degrade gracefully in these mixture scenarios, and is there a lower bound on the contribution from Alice’s LLM required for successful detection?
>
> 2. **Comparison with Learning-based Watermarking Methods**
>    Recent works [1][2][3] have proposed learning-based watermarking methods that embed watermark signals directly into LLMs during training or fine-tuning, enabling downstream detection through LLM outputs. These methods generally require significantly fewer tokens for detection compared to the proposal in this paper.
>
> I found your responses to other reviews intersting. It is clear that several aspects of the paper need to be corrected, revised, expanded with additional details, or even added where content is currently missing. Could you please provide a comprehensive list of the changes you are committing to make in the paper and appendices?
>
> [1] On the Learnability of Watermarks for Language Models.
>
> [2] Watermarking Makes Language Models Radioactive.
>
> [3] Towards Watermarking of Open-Source LLMs.

---

> > ### Author Response · Authors · 2025-08-06
> >
> > We thank the reviewer for their response. We agree that mixing the outputs of Alice's LLM with other (independent) LLMs will make the resulting text harder to detect. But the same is also true for any method for detecting text from a specific language model (i.e., the issue is not specific to our work). We expect with enough tokens our methods should be able to detect mixed-generation text, since a mixture of independent text with text from Alice's model will still exhibit correlation with Alice's data ordering (the independent text can only dilute the memorization effects of the text from Alice's model; **it cannot negate these effects**).
> >
> > In general, learning-based watermarks clearly violate our noninvasiveness criterion (since by definition they require changing the model training process in order to plant the watermark). Regarding the specific references mentioned:
> >
> > [1] requires substantial fine-tuning Alice's model in order to distill an inference-time watermark into the model weights, and is only able to successfully achieve this distillation for inference-time watermarks that induce significant distortion in model output (i.e., watermarks that noticeably change the model's next-token distribution).
> >
> > [2] focuses on a setting where Bob uses the output of Alice's model to train his own model, and assumes Bob's access to Alice's model is mediated by an API through which Alice can choose to apply an inference-time watermark to her model outputs. Their work is inapplicable to our setting.
> >
> > [3] gives a general overview of methods for watermarking open-source/weight language models. These methods violate our noninvasiveness criterion. Moreover, [3] actually draws a fairly negative conclusion about the landscape of existing watermarking methods: "We survey and evaluate existing methods, showing that they are not durable." (This is an excerpt from their paper abstract; by durable, they mean robust to modifications such as quantization, fine-tuning, and model merging). In contrast, at least in the query setting we show our tests are robust to substantial fine-tuning and various other kinds of modifications, including model merging.
> >
> > Regarding changes to the paper, we believe we have addressed the main concerns raised by reviewers either in the existing experiments we mention in our rebuttal or in the additional experiments we have carried out during rebuttal (e.g., experiments with fine-tuned models in the sample setting). We will add the later set of experiments to the final paper. We will also correct all typos pointed out by reviewers, as we acknowledge in our rebuttal. Separately, we have been working on stronger tests for the sample setting, which we hope to include in the final paper. And we have updated various terminology and added exposition for the sake of clarity.
> >
> > It appears that neither this initial review nor follow-up comments included any requests for changes to the paper. If you do have any suggestions—or if we’ve failed to answer any of the questions raised—please let us know, and we’ll be happy to incorporate your feedback!

---

> > > ### Comment · Reviewer_zdZQ · 2025-08-06
> > >
> > > Thank you for your detailed response; most of my concerns are addressed, so I will improve my scores.

---

### Decision · Program_Chairs · 2025-09-17

**Decision:**

Accept (spotlight)

**Comment:**

The paper aims to prove whether a model B is derived (e.g., finetuned) from another model A, and a statistical proof based approach is proposed. The insight behind the paper is that the order of the training samples matters and the model may slightly memorize more the data seen later ("palimpsestic memorization"). Thus, if the original training samples order is saved during the training of model A, they can be used to statistically test whether model B is derived from model A, since further alteration of model A cannot fully erase the "palimpsestic" effect of training sample ordering. Experiments demonstrated the validity and robustness of this approach.

The idea proposed in this paper is novel and is also supported by sufficient experimental results. The contribution of this paper to the field of LLM provenance is significant. Reviewers generally believe that the paper’s quality is high, and additional questions were addressed during the rebuttal. Thus, I recommend the acceptance of this paper. Since the idea in this paper looks quite interesting and may inspire researchers to address other problems in this field, I recommend a spotlight presentation for this paper.